

# Calibrations without raw data—A response to "Seasonal calibration of the end-cretaceous Chicxulub impact event"

Melanie A. D. During[1], Dennis F. A. E. Voeten[1,2], Jeroen (H) J. L. Van der Lubbe[3] and Per E. Ahlberg[1]

[1] Department of Organismal Biology, Uppsala Universitet, Uppsala, Sweden
[2] Natuurmuseum Fryslân, Leeuwarden, Netherlands
[3] Geochemistry & Geology Cluster, Department of Earth Sciences, Vrije Universiteit Amsterdam, Amsterdam, Netherlands

## ABSTRACT

A recent article by DePalma et al. reported that the season of the End-Cretaceous mass extinction was confined to spring/summer on the basis of stable isotope analyses and supplementary observations. An independent study that was concurrently under review reached a similar conclusion using osteohistology and stable isotope analyses. We identified anomalies surrounding the stable isotope analyses reported by DePalma et al. Primary data are not provided, the laboratory where the analyses were performed is not identified, and the methods are insufficiently specified to enable accurate replication. Furthermore, isotopic graphs for carbon and oxygen contain irregularities such as missing data points, duplicate data points, and identical-length error bars for both elements despite different scales, that appear inconsistent with laboratory instrument outputs. A close examination of such methodological omissions and data irregularities can help to raise the standards for future studies of seasonality and prevent inaccurate claims or confirmation bias.

## INTRODUCTION

The End-Cretaceous mass extinction, which included the eradication of the non-avian dinosaurs, is the most widely known of the five big mass extinctions, and is unique in that it was a geologically instantaneous event triggered by a meteorite impact (*Alvarez et al., 1980*; *Smit & Hertogen, 1980*; *Raup, 1986*; *Schulte et al., 2010*). Recently published studies by *DePalma et al. (2021)* and *During et al. (2022)* both attempted to uncover the time of year this impact occurred to improve understanding of the selectivity and severity of the extinction. Both articles conclude that the meteorite impacted in spring (*DePalma et al., 2021*; *During et al., 2022*) or early summer (*DePalma et al., 2021*). However, upon thorough examination of the *DePalma et al. (2021)* article, we have identified anomalies that are unlikely to be the result of analytical work. Portions of this text were previously published as part of a preprint (*During, Voeten & Ahlberg, 2022*).

Corresponding author
Melanie A. D. During,
melanie.during@ebc.uu.se

## STABLE ISOTOPE RECORDS WITH CONFLICTING MIGRATORY SIGNALS

*During et al. (2022)* and *DePalma et al. (2021)* both relied on analyses of stable carbon and oxygen isotope ratios recorded during bone growth of paddlefishes (Polyodontidae) from the Tanis locality in North Dakota (*DePalma et al., 2021*; *During et al., 2022*). The DePalma article also reports stable isotopic data from sturgeons (Acipenseridae) (*DePalma et al., 2021*). However, the number of specimens examined in their study remains unclear, and the article lacks photographs or measurements of the specimens, as well as a reference to the facility where these specimens are stored. This makes it impossible to reproduce their analyses.

Stable isotopic ratios preserved in fossil remains may serve as proxies in reconstructing physiological and environmental conditions and shifts (*Robson et al., 2016*). Carbon isotopic changes in fossil bone can reveal dietary changes and have been used to reconstruct movements of migratory species between marine and freshwater environments (*DeNiro & Epstein, 1978*; *Hobson, 1999*; *Fry, 1981*; *Fry & Sherr, 1989*; *Finlay, 2001*; *Hobson, Piatt & Pitocchelli, 1994*; *Schell, Saupe & Haubenstock, 1989*). Oxygen isotopic changes, typically measured simultaneously, can be used as a palaeoenvironmental proxy for seasonal temperature cycles (*Mizutani et al., 1990*). Furthermore, large $\delta^{18}O$ fluctuations in fossil bone and teeth may also reflect migration between freshwater and marine environments (*Smith et al., 1996*).

Most extant sturgeons are anadromous (*i.e.*, migrate between salt- and freshwater environments) whereas extant paddlefishes are limnetic (*i.e.*, exclusively inhabit freshwater habitats) (*Robson et al., 2016*; *Bemis & Kynard, 1997*; *LeBreton, Beamish & McKinley, 2004*). The oxygen isotopic graphs of sturgeon bone in *DePalma et al. (2021)* exhibit large annual variations, presumably reflecting the anadromous nature of these fishes, whereas the corresponding graphs for paddlefish bone register little variation (*DePalma et al., 2021*). Despite these contrasting migratory strategies recorded in $\delta^{18}O$, the carbon isotopic graphs for the sturgeons and paddlefishes are essentially identical (*DePalma et al., 2021*); there is no signal difference that can be attributed to anadromous and limnetic lifestyles (*Robson et al., 2016*; *DeNiro & Epstein, 1978*; *Hobson, 1999*; *Fry, 1981*; *Fry & Sherr, 1989*; *Finlay, 2001*; *Hobson, Piatt & Pitocchelli, 1994*; *Schell, Saupe & Haubenstock, 1989*). The presented data would imply that limnetic paddlefishes and anadromous sturgeons maintained remarkably similar diets despite their contrasting migratory strategies. While the observed $\delta^{13}C$ cyclicity in sturgeon bone remains within the plausible value range for the marine realm, its near-perfect alignment with the plot trajectories in the paddlefish records is highly unexpected. Although the exact relationship between $\delta^{18}O$ and $\delta^{13}C$ in Tanis acipenseriform fossils is only now described for the first time, rendering background information understandably limited, this surprising correlation warrants further explanation.

### Primary data

No primary isotopic data are openly shared, either with the article itself or through a linked online repository. The publication also lacks a Data Availability Statement

(*DePalma et al., 2021*), despite such a statement being a requirement for publication in *Scientific Reports*.

## Analytical facility

The facility where the stable isotopic analyses were carried out is not revealed, nor are the dates when the experiments were conducted. Curtis McKinney, who is stated to have carried out the analyses (*DePalma et al., 2021*), passed away in 2017 and cannot be consulted about the referred analytical work.

## Methods

The isotopic section of the Methods declaration is brief and lacks a sufficiently detailed account of the techniques, sample weights, and adopted standards necessary for replicating the analyses. To adhere to best practices in reporting, the following methodological details should be provided:

- Description of the analytical facilities used and the specific location of the laboratory where the analyses were conducted;
- Estimation of sample weights (in milligrams or micrograms) after extraction;
- Description of sample treatment, including the acids and bases used, as well as the procedures followed;
- Identification of carbonate standards utilised for calibration purposes;
- Specification of the analytical precision achieved.

These methodological details are conventionally declared in published literature, as evidenced by studies cited (*Schulte et al., 2010*; *During et al., 2022*; *During, Voeten & Ahlberg, 2022*; *Robson et al., 2016*; *DeNiro & Epstein, 1978*; *Hobson, 1999*; *Fry, 1981*; *Fry & Sherr, 1989*; *Finlay, 2001*; *Hobson, Piatt & Pitocchelli, 1994*; *Schell, Saupe & Haubenstock, 1989*; *Mizutani et al., 1990*; *LeBreton, Beamish & McKinley, 2004*; *Brand et al., 2014*; *Cullen et al., 2019*; *De Rooij et al., 2022*; *Coplen, 2011*; *Bryant et al., 1996*; *Vennemann et al., 2001*; *Hodgkins et al., 2020*; *Passey et al., 2005*; *Breitenbach & Bernasconi, 2011*; *Torres, Mix & Rugh, 2005*).

## Sampling density and amount of carbon

Stable isotope graphs in *DePalma et al. (2021)* imply that, in some cases, up to 43 samples must have been successfully obtained along an 800 μm transect. Since the drill bits used for sampling are conical in shape, drilling deeper to yield more material inevitably also widens the drill holes. Caution thus needs to be exercised as to avoid intersection of the drill holes and prevent contaminated spot sampling. A total of 43 individual samples yielded along an 800-μm-long transect corresponds with a maximum drill bit width of 18.60 μm, which equates to much less than half the diameter of a typical human hair. Only 10% by weight of hydroxyapatite consists of the structural carbonate (*Torres, Mix & Rugh, 2005*) that is available for stable isotope analysis. Since the amount of material required for reliable stable isotope analysis varies across the potentially available analytical techniques, the protocols used for obtaining sufficient material as well as for conducting isotopic analyses

at the reported spatial resolution require explicit specification. Although the reported resolution may have theoretically been achievable prior to 2017, when the co-author declared responsible for the analyses passed away, it must always have involved a novel approach or modification from an existing protocol. The parameters involved require a declaration to ensure reproducibility, but such a specification is lacking here (*DePalma et al., 2021*).

## Graphs in the article and in the Supplementary Information

Figure 2 of the *DePalma et al. (2021)* contains the stable isotope record (43 sampling spots) and osteohistology of specimen number FAU.DGS.ND.755.57.T, which is a sturgeon pectoral fin spine. However, this exact specimen number is also declared in the supplemental information of *DePalma et al. (2021)* associated with a different graph that only involves 35 sampling spots. That graph also includes a line dip without a marked data point, which may be a 36th sampling point, but this has not been clarified. Multiple inconsistencies are best explained as graphs (*DePalma et al., 2021*) that were produced by image-handling software. Since the article does not provide the original data (*DePalma et al., 2021*), these results cannot be corroborated or understood. The referred issues described below are also illustrated in Figs. 1–9 of this article.

According to the figures in *DePalma et al. (2021)*, each sample location yielded hydroxyapatite samples for stable carbon and oxygen isotope analyses. The methods (*DePalma et al., 2021*) declare that these samples were analysed using a Gas Bench II linked to a Thermo Finnigan dual-inlet MAT 253 Stable Isotope Ratio Mass Spectrometer (*DePalma et al., 2019*). The analytical setup in this study (*DePalma et al., 2021*) mirrors the one utilised at the Vrije Universiteit Amsterdam (*During et al., 2022*). However, our experience (*During et al., 2022*) indicates that this configuration does not allow for the high-resolution sampling of such small samples (*DePalma et al., 2021*), unless additional measures are taken, such as temporarily cryo-focussing of the produced $CO_2$ with liquid nitrogen (*During et al., 2022*). Notably, such an approach was not described or declared in *DePalma et al. (2021)*.

Carbonate, in this case as a component of the hydroxyapatite samples, undergoes a reaction with orthophosphoric acid ($H_3PO_4$), leading to the generation of $CO_2$ gas (*Torres, Mix & Rugh, 2005*). Subsequently, this sample gas is introduced into the mass spectrometer, where simultaneous measurements are taken for stable carbon and oxygen isotope ratios (*Torres, Mix & Rugh, 2005*). Consequently, a single measurement provides ratios between $^{12}C$ and $^{13}C$, as well as between $^{16}O$ and $^{18}O$, derived from the $CO_2$ gas within a single sample. The expected outcome is a data table capturing both carbon and oxygen isotopic ratios for each sampling point (*Torres, Mix & Rugh, 2005*).

Graphing software is then employed to generate two spot values, one for oxygen and one for carbon, precisely aligned on the same vertical line (*Breitenbach & Bernasconi, 2011*). However, it is important to recognize that analyses may face challenges due to various factors. Failures could stem from a breach of vacuum or low-amplitude measurements, compromising the reliability of both carbon and oxygen measurements. Alternatively, if the analysis of either carbon or oxygen fails, such as due to high inter-peak

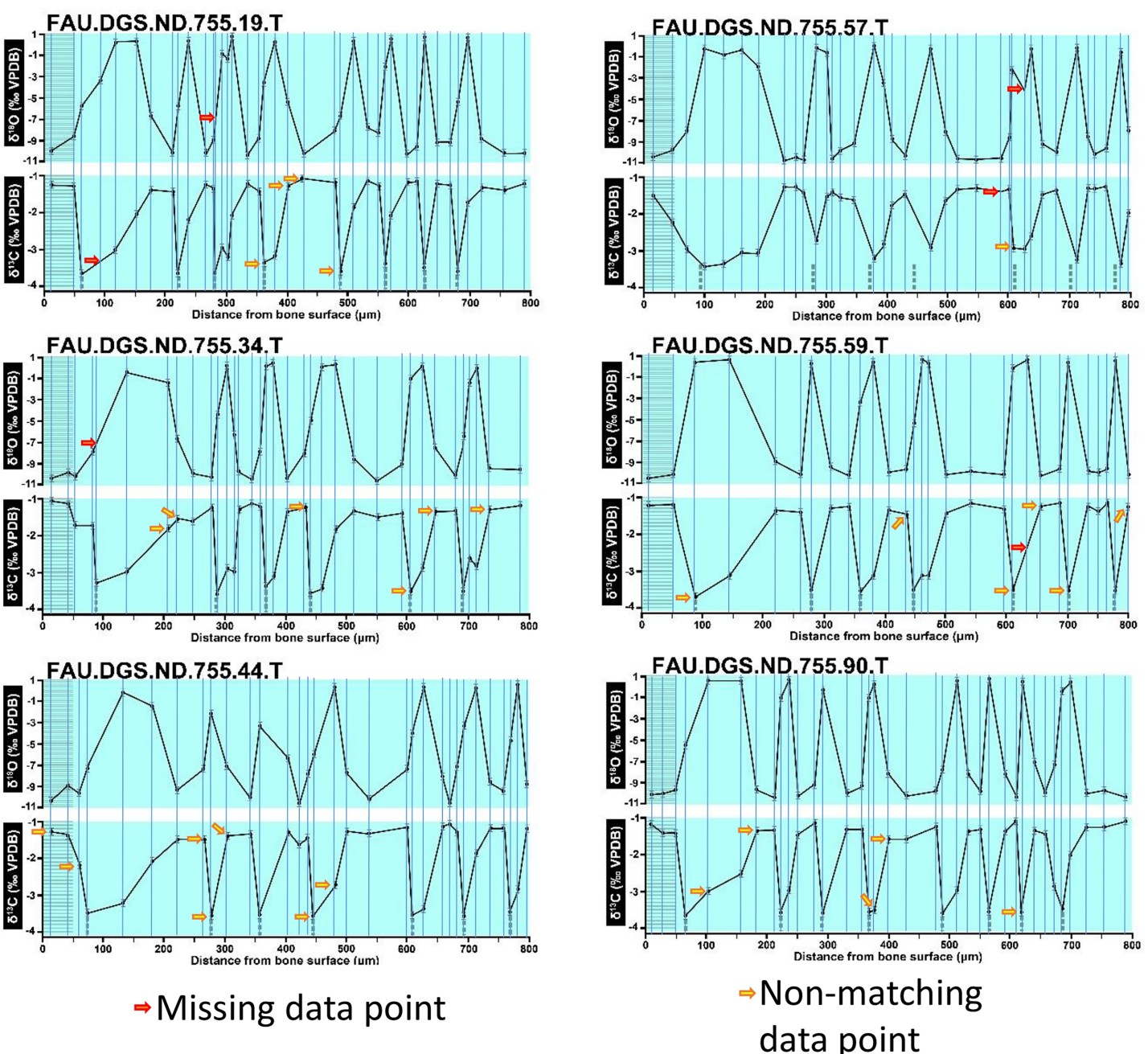

**Figure 1** **Missing and misaligned data points in SUP MAT 9 (*DePalma et al., 2021*).** Top right is sturgeon specimen FAU.DGS.ND.755.57.T, the same specimen number as Figure 2 (*DePalma et al., 2021*; (see also Figure 5 in this article)). Figure modified from *DePalma et al. (2021)* (CC BY 4.0).                                                            

variation in the measurement of either element, the other element can still be considered reliable and plotted independently.

It is noteworthy that only $^{2}/_{3}$ of the oxygen from carbonate ($CO_3^{2-}$) is converted into the $CO_2$ that can be measured by the isotope ratio mass spectrometer. Consequently, the

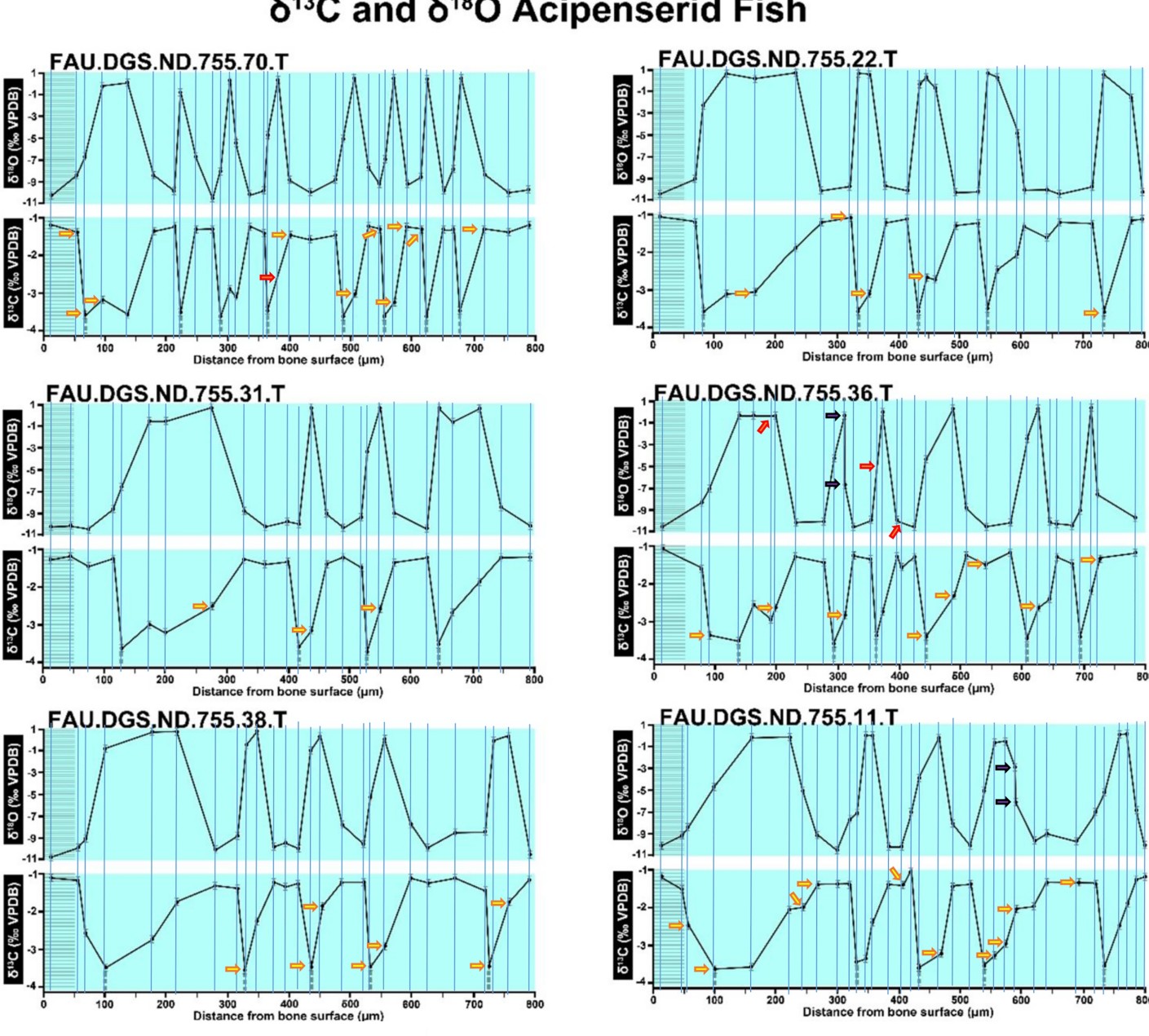

**Figure 2 Missing, duplicate, and misaligned data points in SUP MAT 10 (*DePalma et al., 2021*).** Figure modified from *DePalma et al. (2021)* (CC BY 4.0).

oxygen isotope values are highly sensitive to conditions during acid digestion, particularly temperature. Interestingly, some samples omit both carbon and oxygen isotopic values as data points (*DePalma et al., 2021*). When isotopic values of one of the elements are

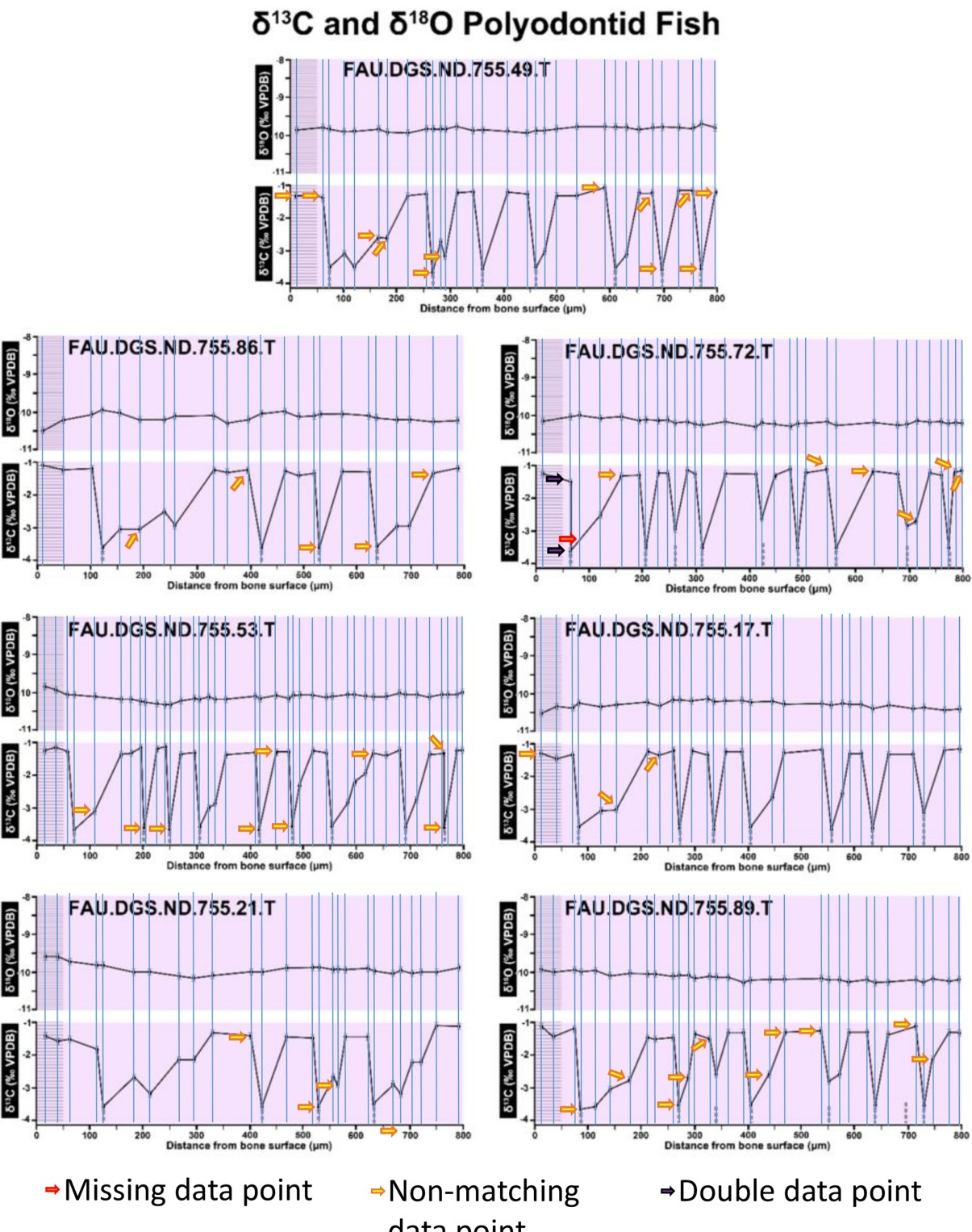

**Figure 3  Missing, duplicate, and misaligned data points in SUP MAT 11 (*DePalma et al., 2021*).** Figure modified from *DePalma et al. (2021)* (CC BY 4.0).

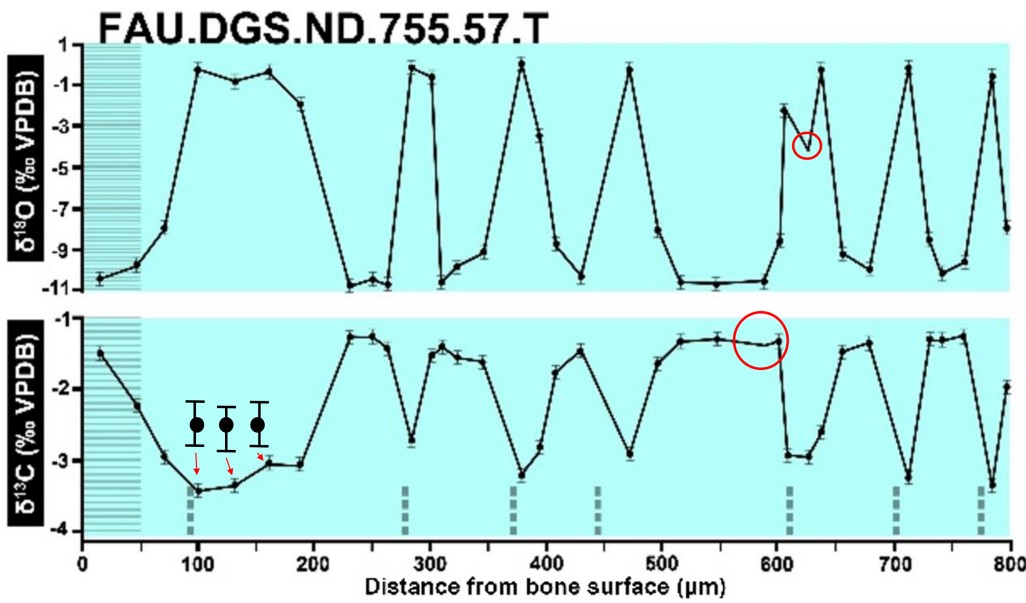

**Figure 4** The top right graph of Sup Mat 9 (*DePalma et al., 2021*). Dips without sampling points and uncentered error bars. Figure modified from *DePalma et al. (2021)* (CC BY 4.0).

omitted, it is advisable to at least provide a threshold value and a possible explanation for these failures.

In the graphs presented in the SUP MAT 9-11 (*DePalma et al., 2021*), we observe multiple instances where only one of the two isotope measurements is displayed, and in many cases, the two spots are not perfectly vertically aligned (see Figs. 1–3 in this article). The misalignment observed in these graphs suggests that they may not be based on isotopic measurements from the same $CO_2$-molecule. Alternatively, they were not produced by graphing software. Conversely, in three places we encounter double and widely separated measurements of one of the two isotope ratios on the same vertical line. Double measurements of a single isotope, visualising both a minimum and maximum in their range, supposedly from a single sample point, as depicted in our Figs. 2 and 3, cannot be attributed to technical readouts. Furthermore, in the top right graph of the Supplemental Materials 9 of *DePalma et al. (2021)* (see our Fig. 4), the line graph exhibits two noticeable 'dips' without corresponding sampling points or error bars, and no explanation for these anomalies is provided. The error bars accompanying the carbon isotope records in Fig. 2 of the article (*DePalma et al., 2021*) (Fig. 5 in this article) differ fundamentally from all the error bars in the carbon isotope records in the supplemental information of *DePalma et al. (2021)*, which seem to be duplicates of the oxygen isotope error bars. Despite the considerable difference in the vertical scale of the graphs, the error bars on both the oxygen and carbon plots are identical in length. Although the Methods section (*DePalma et al., 2021*), specifies a precision of ±0.3‰, this level of precision is not reflected in the graphs. The error bars are often not centred correctly and display inconsistencies, deviating from the uniform values and appearance expected of analytical software (Figs. 5 and 6 of this article). Moreover, as no carbonate or apatite standard has

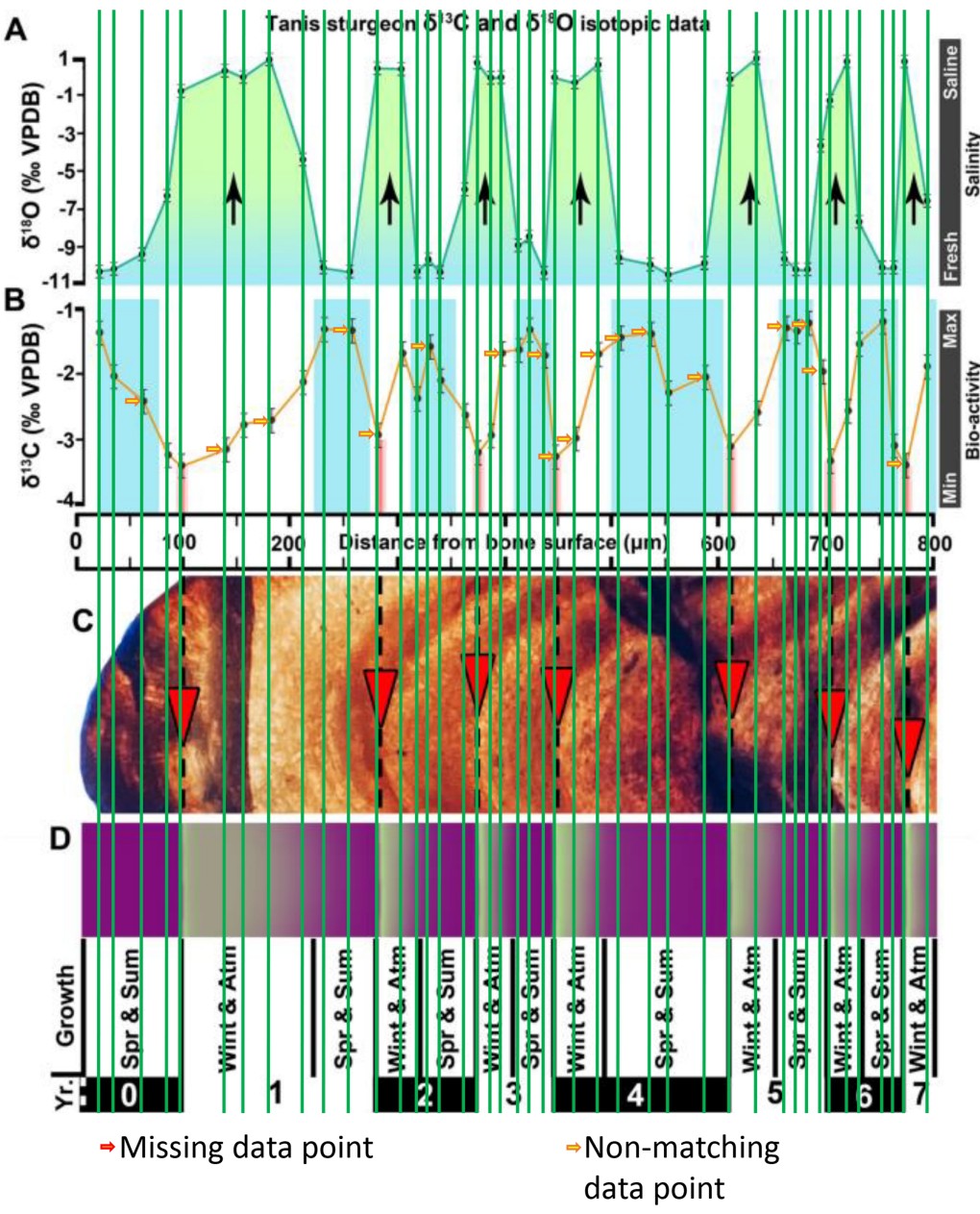

**Figure 5 Misaligned data points of Figure 2 of *DePalma et al. (2021)*.** The sturgeon specimen FAU. DGS.ND.755.57T is the same specimen number as in the top right of SUP MAT 9 (*DePalma et al., 2021*) (see also Fig. 1 in this article). The histological section (C) is out of focus, the osteocyte distribution cannot be assessed here. Figure modified from *DePalma et al. (2021)* (CC BY 4.0).

been specified, the origin of the error estimate remains unexplained. The first author of the article stated that these error bars represent icons (communicated in peer-review) that have been misinterpreted as error bars. This of course may explain their homogeneity, but it is recommended to plot actual error bars in graphs of published articles.

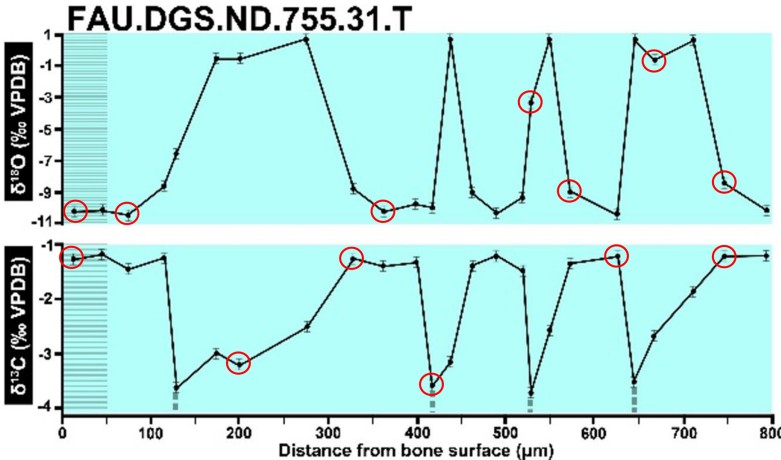

**Figure 6 The left, centre, graph of Sup Mat10 (*DePalma et al., 2021*).** Incomplete error bars. Figure modified from *DePalma et al. (2021)* (CC BY 4.0).     

We interpret these characteristics to be inconsistent with direct outputs of analytical graphing software. Moreover, they cannot represent faithful reproductions of machine-generated analytical graphs, as they exhibit features that are inconsistent with copying errors. These features notably include the double measurements of an isotope and use of icons that resemble error bars but do not represent true error estimates. The pattern between the oxygen and carbon value maxima and minima across all referred specimens is extraordinarily consistent (Figs. 1–6 of this article). Achieving such consistency would necessitate perfect sampling of growth intervals without the mixing of bounding layers. Even under ideal conditions, sample heterogeneity as well as analytical uncertainty (*Breitenbach & Bernasconi, 2011*) will generally prevent reproduction of these exact values in the same individual—let alone in others. Notably, the stable isotope curves consistently exhibit equivalent maximum and minimum values each year in each individual fish, implying a consistent annual diet, a pattern not observed in any comparable analysis known to us.

The overlay image presented in SUP MAT 12 (*DePalma et al., 2021*) exemplifies the problem. It provides no scale on the x-axis, but verification against one of the source figures, SUP MAT 10 (*DePalma et al., 2021*) (Fig. 7 in this article), which has such scale bars, reveals features that require explanation. The curves for specimens. FAU.DGS. ND.755.38.T., FAU.DGS.ND.755.22.T and FAU.DGS.ND.755.11.T match almost perfectly when overlaid: over the course of the eight recorded growing seasons, it shows all three individuals depositing exactly 800 µm of bone. While matching sequences of "fat years" and "lean years" are plausible and can be expected, as they represent responses to the same ecosystem that hosted all the individuals, this perfect match of total amount of bone deposition is remarkable. Furthermore, the curves of FAU.DGS.ND.755.70.T appear at first sight not to match those of the three aforementioned specimens, but in fact do so perfectly when aligned to the left margin and stretched rightwards to exactly 150% of original length (Fig. 8 of this article). Similarly, the curves of FAU.DGS.ND.755.31.T fit perfectly onto those of FAU.DGS.ND.755.36.T when aligned to the left margin and

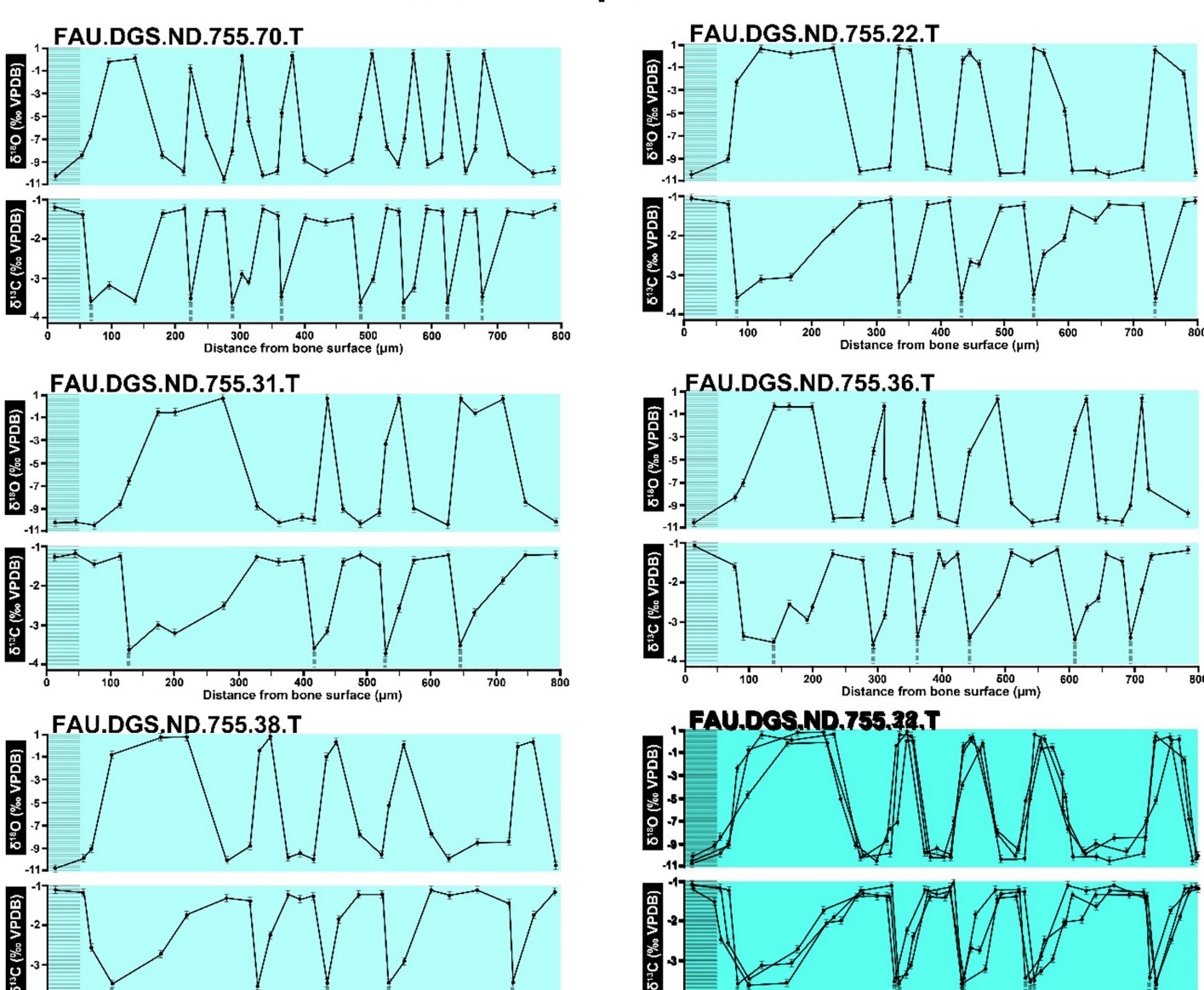

# δ¹³C and δ¹⁸O Acipenserid Fish

**Figure 7 Overlay of SUP MAT 10 (*DePalma et al., 2021*) FAU.DGS.ND.755.38.T. (bottom left) and FAU.DGS.ND.755.11.T (bottom right).** Figure modified from *DePalma et al. (2021)* (CC BY 4.0).

shortened to exactly 70% of original length (Fig. 9 of this article). It seems improbable for these specimens to have been growing consistently at rates of 150% faster and 70% slower, necessitating an explanation to confirm the validity of the data.

## THIN SECTIONS IN THE SUPPLEMENTARY MATERIALS

Upon closer examination of the microscope images provided in the supplementary materials of the *DePalma et al. (2021)*, we identified a noteworthy discrepancy. Specifically,

# δ¹³C and δ¹⁸O Acipenserid Fish

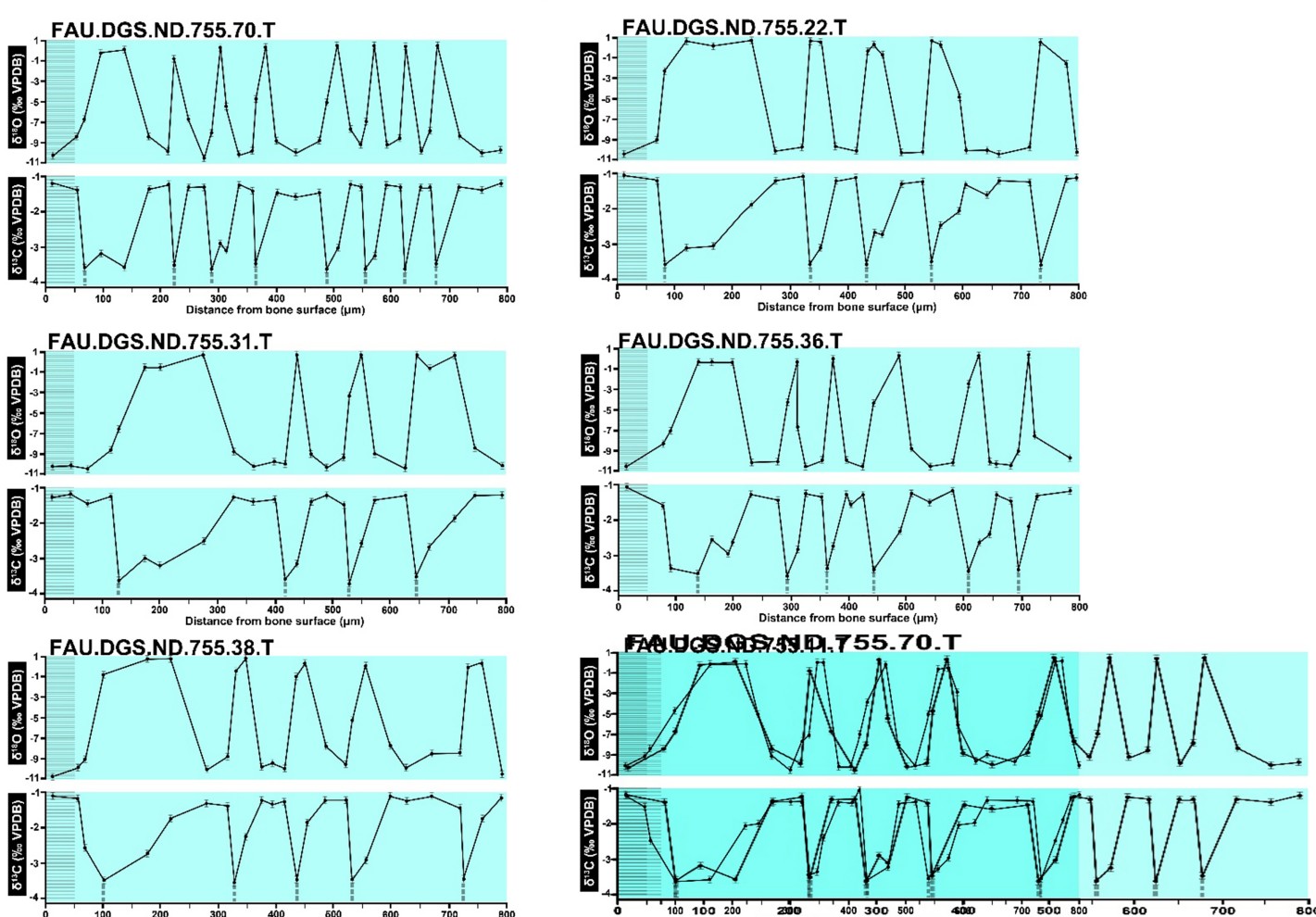

**Figure 8 Overlay of SUP MAT 10 (*DePalma et al., 2021*) FAU.DGS.ND.755.70.T (top left), stretched to 150% of original length, onto FAU. DGS.ND.755.11.T (bottom right).** Figure modified from *DePalma et al. (2021)* (CC BY 4.0).   

SUP MAT 3 (*DePalma et al., 2021*) SUP MAT 5 (*DePalma et al., 2021*) depict what appears to be the same section, albeit different photographs of which one is horizontally flipped (see Figs. 10–12). In SUP MAT 3, the specimen in question is identified as FAU.DGS. ND.755.36.T, while in SUP MAT 5, it is labelled as FAU.DGS.ND.755.57.T. Although the two images appear to have been captured under different conditions—likely with one photographed through a microscope operated with a digital camera, and the other photographed through the microscope's ocular using a handheld camera or phone.

Both images contain an air bubble, a phenomenon typically observed on the rear of a histological slide beneath the protective glass cover (see Fig. 12). In our experience, a histological slide is glued on a protective glass cover, and one prefers to examine the opposite, uncovered, side for analysis.

The use of different capture methods has introduced minor deviations between the images, specifically geometric distortion in one of the photographs. SUP MAT 5, which

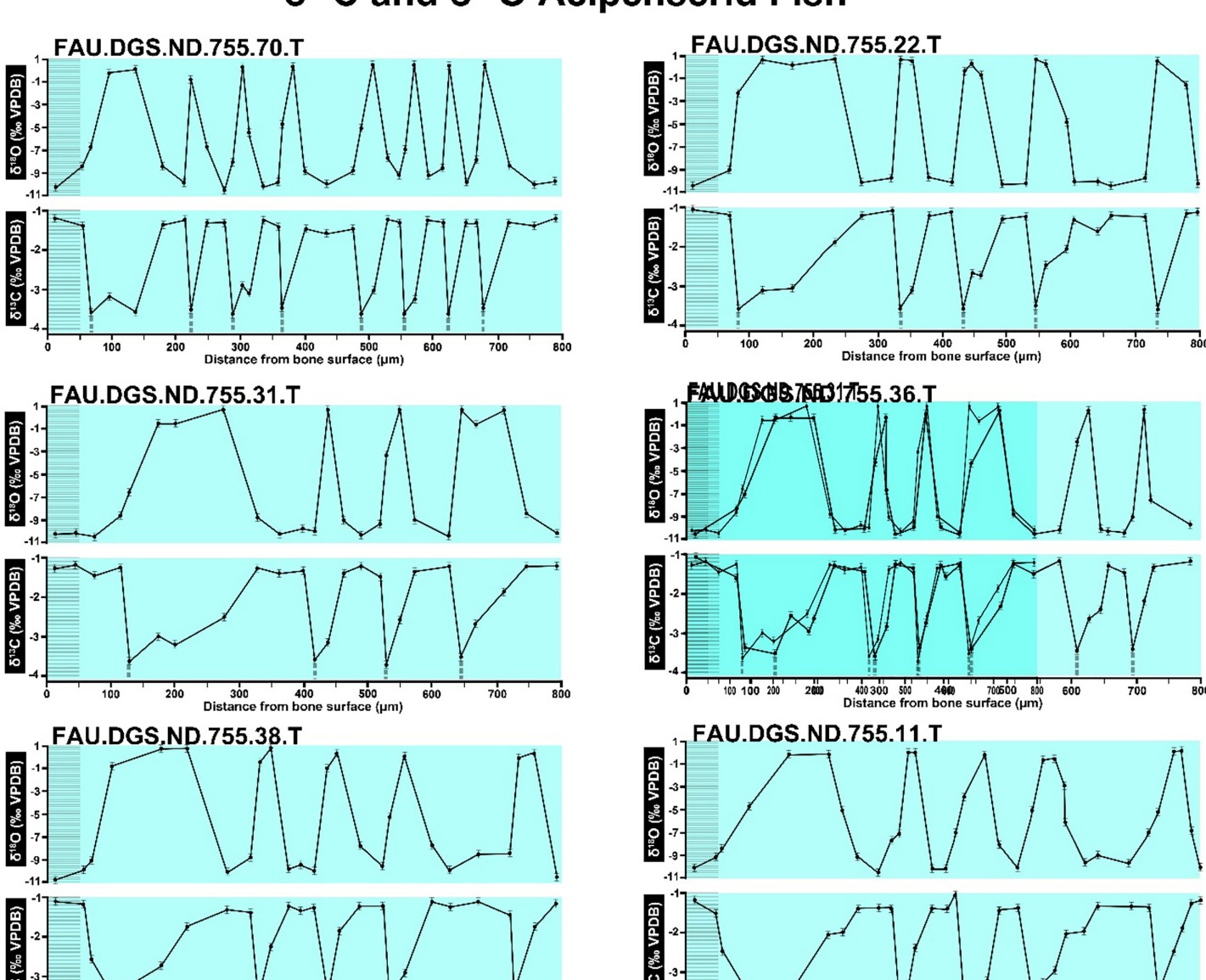

**Figure 9 Overlay of SUP MAT 10 (*DePalma et al., 2021*) FAU.DGS.ND.755.31.T (left, centred), compressed to 70% of original length, onto FAU.DGS.ND.755.36.T (right centred).** Figure modified from *DePalma et al. (2021)* (CC BY 4.0).

was likely captured using a calibrated camera mounted to the microscope, does not exhibit such distortion. In contrast, SUP MAT 3 shows distortion, likely caused either by the lens of a handheld camera or from photographing through the microscope's ocular. This type of distortion (*Lucchese & Mitra, 2003*) is well-documented and occurs when calibration is not applied. After compensating for the geometric distortion in SUP MAT 3 (See Fig. 13), all features align perfectly, confirming that both images depict the same section of the
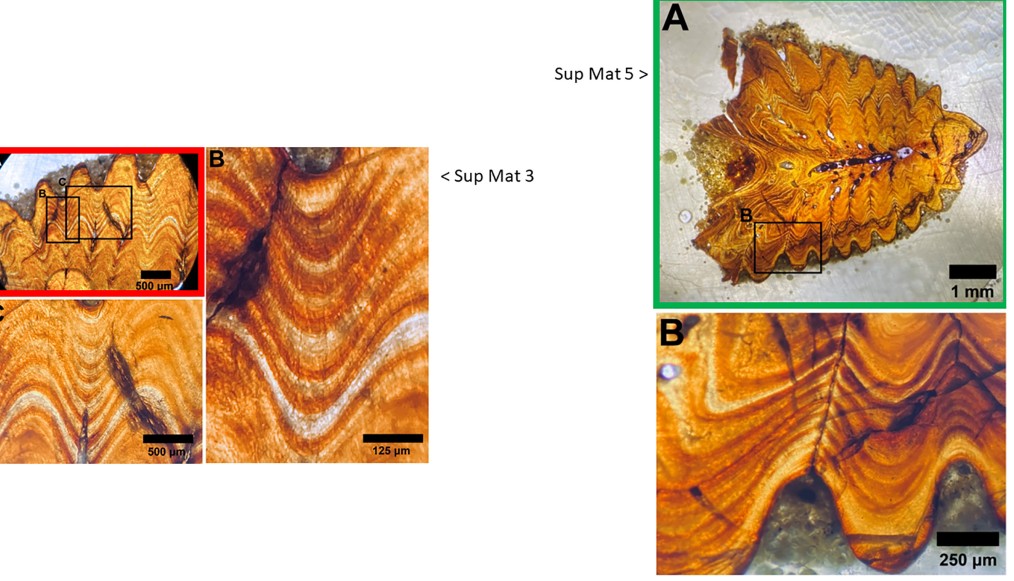

**Figure 10  SUP MAT 3 (*DePalma et al., 2021*) (left) and SUP MAT 5 (*DePalma et al., 2021*) (right).** Figure modified from *DePalma et al. (2021)* (CC BY 4.0).

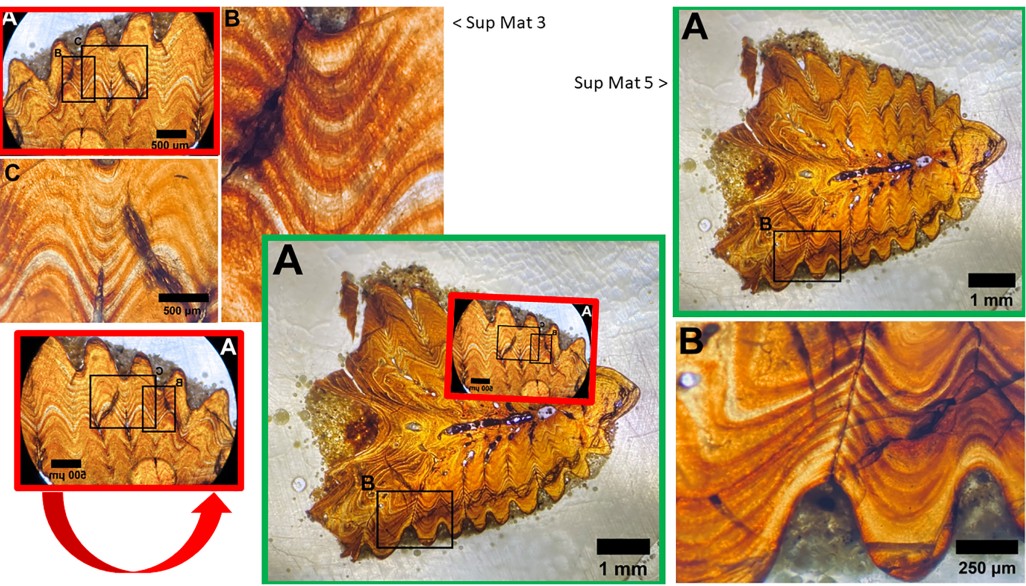

**Figure 11  Juxtaposition of SUP MAT 3 (*DePalma et al., 2021*) (left, red) and SUP MAT 5 (*DePalma et al., 2021*) (right, green; see also Fig. 10).** When flipped horizontally (bottom left), the morphology and histology of the sample section in SUP MAT 3 (*DePalma et al., 2021*) matches that in SUP MAT 5 (*DePalma et al., 2021*) exactly. This is demonstrated (centre, bottom) where SUP MAT 3 (*DePalma et al., 2021*) is superimposed on SUP MAT 5 (*DePalma et al., 2021*). Figure modified from *DePalma et al. (2021)* (CC BY 4.0).

specimen. The images share identical features, such as vascular canals, cracks, LAGs, and an air bubble in identical locations.

   This observation suggests that the images are of the same histological section, but for unexplained reasons, one has been flipped and both were attributed to different specimens.

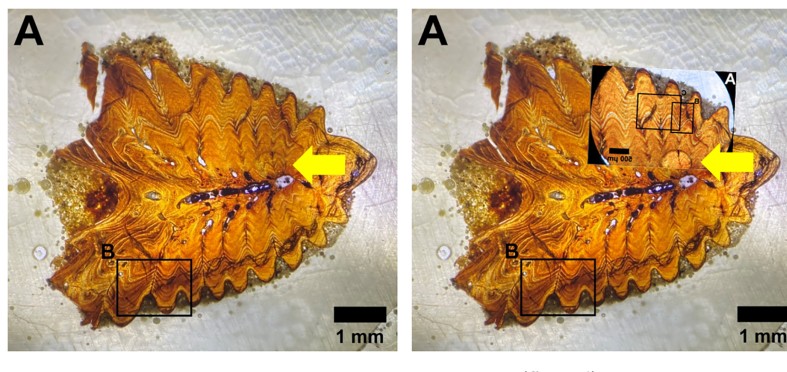

Sup Mat 5            Sup Mat 3 (flipped) + Sup Mat 5

**Figure 12** **Left: SUP MAT 5 (*DePalma et al., 2021*) with a faint air bubble (yellow arrow); Right: SUP MAT 3 (*DePalma et al., 2021*) overlying SUP MAT 5, with a duplicate air bubble in exactly the same position (yellow arrow).** Furthermore, the air bubble is in focus but the osteohistology is not, rendering the assessment of the seasonal time of death from this visualisation impossible. Figure modified from *DePalma et al. (2021)* (CC BY 4.0).

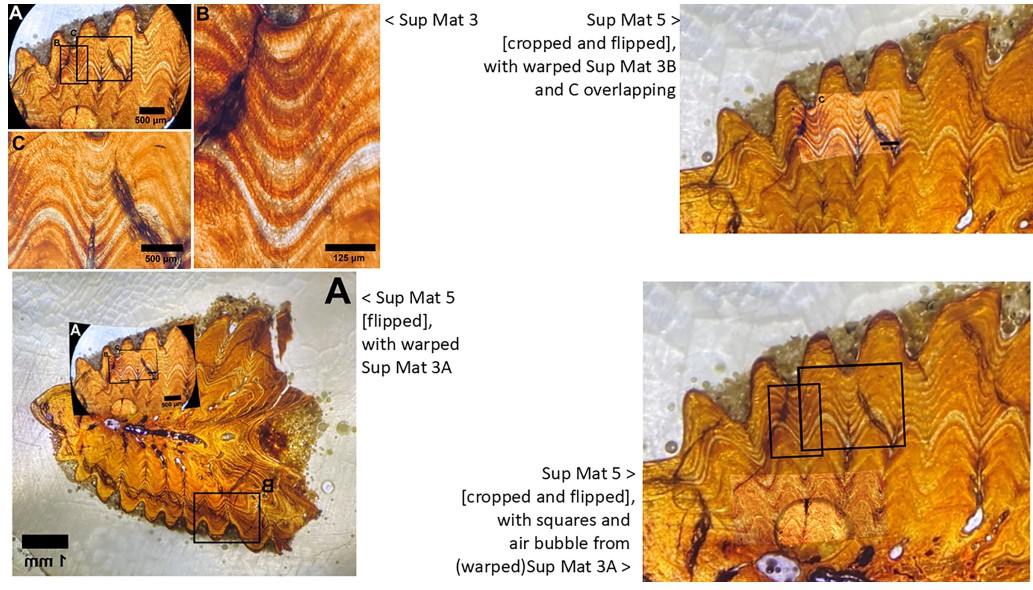

**Figure 13** **Sup Mat 3 (warped) corresponds precisely with (flipped) Sup Mat 5.** Left: Top: SUP MAT 3 (*DePalma et al., 2021*) with marked areas (A, B and C). Bottom: SUP MAT 5 (*DePalma et al., 2021*), overlaid with a geometrically corrected (warped) version of the same region from SUP MAT 3A. Right: Top: Cropped portions of SUP MAT 3B and 3C overlaid on SUP MAT 5, revealing a perfect alignment of all key features. Bottom: SUP MAT 5 showing indicator squares corresponding to SUP MAT 3B and 3C, alongside a warped overlay of SUP MAT 3A's air bubble. After compensating for the geometric distortion, all visible features, such as the air bubble, growth bands, and vascular canals, match perfectly between SUP MAT 3A and SUP MAT 5. Figure modified from *DePalma et al. (2021)* (CC BY 4.0).

Although one could potentially explain this by having accidentally submitted the wrong file in the Manuscript Central figure submission slot, no such explanation has been provided.

## Fish sizes

Figure 3 (*DePalma et al., 2021*) illustrates the sizes of "sub-yearling" fishes. In this figure, the number of measured fishes is unspecified, and a corresponding data table providing individual fish measurements is not included. Upon revisiting the Supplementary Information of the article describing the locality (*Torres, Mix & Rugh, 2005*), a contradictory graph is shown, suggesting that fish sizes commence at 15 cm. However, in *DePalma et al. (2019)*, raw data and a description of the measurement protocol are lacking.

   *DePalma et al. (2021)* state that "The smallest Acipenseriformes at Tanis (<16 cm fork length) fall below the expected length of yearling extant acipenseriform taxa, and we interpret that they died during sub-yearling ontogeny (*i.e.*, YOY)." This contradicts the depiction of fish sizes in the initial Tanis article (*Lucchese & Mitra, 2003*), where all fishes are described as being at least 15 cm in length. Additionally, the "Tanis fossil sizes" range indicated in Fig. 3 (*DePalma et al., 2021*) suggests that the fishes measure between 8–12 cm in length.

   Moreover, there is a lack of explanation for the data points in Fig. 3 (*DePalma et al., 2021*), with several instances of fish with the same body size allocated to different 'spawning seasons'. While Fig. 1 (*DePalma et al., 2021*) does include depictions of juvenile fishes, inconsistencies in the scale bars prevent accurate size reconstruction, raising concerns about the reliability of these size claims. For example, the width of section "E" in Fig. 1A is roughly one third of the width of the 3 cm scale bar in the same figure, which makes it impossible for "E" to have a valid 1 cm scale bar that spans less than half its own width. Additionally, the fishes depicted in Fig. 1 (*DePalma et al., 2021*) do not significantly differ in size from the other works describing fishes extracted from Tanis by Steve Niklas and Robert Coleman in 2010 (*Hilton et al., 2023*; *Hilton & Grande, 2023*), which do not mention any small fishes that could be considered hatchlings. During fieldwork in 2017, no sub-yearling fish was observed, and body sizes were not measured or recorded (personal observation M. During, 2017).

## Conclusions of a spring death

The histological sections shown in *DePalma et al. (2021)* contain air bubbles, and the histology itself appears out of focus, indicating that they may have been photographed from the side of the protective glass cover rather than from the side of the sample. Consequently, it becomes challenging to properly assess the histology of the specimens, making it difficult to discern osteocyte distribution (*Castanet, 1993*; *Davesne et al., 2020*; *De Buffrénil, Quilhac & Castanet, 2021*). The approach selected by *DePalma et al. (2021)* instead focused on the "darker" and "lighter bands". This represents a more basic approach that is sometimes used in age-at-death estimates of vertebrates (*Brennan & Cailliet, 1989*). By binarizing a histological section into either "lighter bands" (associated with autumn and winter) and "darker bands" (associated with spring and summer), one can indeed discriminate between death in autumn/winter or spring/summer. However, quantification of osteocyte distribution is more reliable and allows for distinction between the four

individual seasons (*During et al., 2022*; *Davesne et al., 2020*). Furthermore, the low resolution of the histological sections and the modifications in the graphs do not meet the standards for publication in *Scientific Reports*.

While the stable carbon isotope record displays oscillations between summer maxima and winter minima, akin to findings in *During et al. (2022)*, the final measurements of all specimens in *DePalma et al. (2021)* exhibit a maximum, suggesting fishes that perished not in spring but rather in late summer, similar to the previous maxima that are rapidly followed by a dip to the minima. *DePalma et al. (2021)* conclude that these fishes perished in late spring early summer, but irregularities in the methods and data presented appear sufficient to undermine confidence in either conclusion.

## CONCLUSIONS

The stable isotope graphs presented in *DePalma et al.*'s *(2021)* article, as depicted in Fig. 2 (*DePalma et al., 2021*) and the supplemental materials of *DePalma et al. (2021)*, exhibit patterns that deviate from what would be expected from direct analytical outputs. Notably, neither the raw isotope data nor the isotopic results of the measured standards are provided in the article or in any linked repository. Additionally, the supplemental materials of *DePalma et al. (2021)* contains photographs of the same specimen taken with two different cameras, one of which was flipped and attributed a different specimen number. Addressing these anomalies requires an explanation and a detailed account of the analytical procedures, including laboratory protocols, sample weights, standards utilised, and micro-milling transects, among other aspects.

Crucially, in accordance with *Scientific Reports'* standards, the authors must provide the raw data underpinning their published figures. We strongly recommend that the Editor in Chief investigate how an analytical article was published in *Scientific Reports* without the mandatory Data Availability statement.

Addressing these issues is imperative to uphold the integrity and transparency of scientific research.

## ACKNOWLEDGEMENTS

We would like to thank Jan Smit for his advice, and we would also like to thank the editors and reviewers (Thomas Cullen and Robert DePalma) at PeerJ for their insightful feedback that greatly helped improve our work.

### Funding
The authors received no funding for this work.

### Competing Interests
The authors declare that they have no competing interests.
## Author Contributions

- Melanie A. D. During conceived and designed the experiments, performed the experiments, analyzed the data, prepared figures and/or tables, authored or reviewed drafts of the article, and approved the final draft.
- Dennis F. A. E. Voeten performed the experiments, analyzed the data, authored or reviewed drafts of the article, and approved the final draft.
- Jeroen (H) J. L. Van der Lubbe conceived and designed the experiments, performed the experiments, analyzed the data, authored or reviewed drafts of the article, and approved the final draft.
- Per E. Ahlberg conceived and designed the experiments, performed the experiments, analyzed the data, prepared figures and/or tables, authored or reviewed drafts of the article, and approved the final draft.

## Data Availability

This is a reanalysis of the dataset published in DePalma, R.A., Oleinik, A.A., Gurche, L. P. et al. Seasonal calibration of the end-cretaceous Chicxulub impact event. Sci Rep 11, 23704 (2021). https://doi.org/10.1038/s41598-021-03232-9.

## Supplemental Information

Supplemental information for this article can be found online at http://dx.doi.org/10.7717/peerj.18519#supplemental-information.

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
