# Peer review of "Calibrations without raw data—A response to "Seasonal calibration of the end-cretaceous Chicxulub impact event""

_PeerJ, doi:10.7717/peerj.18519_

## Round 0.1 · original submission · Major Revisions

You make some interesting points about a previously published paper and some good practice recommendations on how to avoid them. The preprinted manuscript was already reviewed and deemed valid by a Peer Community in Paleontology editor and two reviewers and normally could be published like this in PeerJ. However, as it concerns a response to an existing article then, according to our policy – the original author(s) should also be invited to submit a signed review and valid points of critique must be addressed. I also invited additional reviewers not directly involved with the original paper and who have not published or been involved with the authors of this work.

The paper addresses points raised related to the original published paper. As far as I can verify most points seem still valid and are mostly concerned with lack of raw data making that some of results reported by DePalma et al. remain irregular, difficult to reproduce and unverifiable given the materials provided with the original study. Despite, an Editor's Note published (09/12/2022) which reads “Readers are alerted that the reliability of data presented in this manuscript is currently in question. Appropriate editorial action will be taken once this matter is resolved.”, most points remain valid including lack of full publicly available primary isotopic data being shared in the paper or publicly anywhere at the time of checking (7 May 2024). However, some additional points are raised which need further documentation or support. Some points need to be removed or at minimum rephrased due to lack of evidence or possible alternative explanations which are currently not entertained (e.g., suspicion of figure manipulation without evidence, appraisal of supplementary images without acknowledging alternative explanations)

I agree with reviewer 2 (the author of the work being critiqued) that a reply/rebuttal should focus on the correct representation of published facts. It should not include allegations/interpretations based on unverifiable claims or (closed) personal investigations or opinions.

However, the mentioned report of the investigation by the University of Manchester does confirm that mistakes were made and at least some of the potential implications are listed in this rebuttal. I therefore would like the authors to focus on the scientific points raised to the original work rather than investigations or difficult-to-verify claims.

Irrespective of the reasons behind them, I agree with reviewer 1 that any of the points, if they can be substantiated, raise some questions about the editorial and peer-review process (as well as the formal rebuttal article process) and the data availability and transparency.

The eight points raised by Reviewer 2 as reasons to reject the publication of the manuscript are in my opinion, not sufficient to warrant rejection (the reviewer’s comments are in [brackets] and my reply to each point is designated by >>>):

1) [Relationship to a finished investigation] >>> The issues raised in the reply pertain to the original publication and seem valid as demonstrated least by reviews of two external reviewers and the editor at PCI and are, in that sense, independent of allegations. The paper is written as a reply to the original published materials and not the complaint. It is true that the results of the investigation of the complaint are published but we do not have access to the original documentation, and neither is it published as peer-reviewed text with a digital object identifier. At least some of the issues raised in the reply were confirmed and even when the reason behind them is revealed, it does not change the fact that they are still perpetuated in the published and currently available work. Nowhere in the reply, is there mention of data fabrication, scooping or corruption which would be serious allegations. There is however a claim of figure manipulation (line 249-250) which is not clearly substantiated and hard to prove as it may have various alternative explanations. Neither does the reply seem to be a smear campaign as it highlights issues based on the originally published and still available materials and, particularly, associated isotope data which are irregular, as stated by external reviewers and the handling recommender of PCI and therefore they are not necessarily a reason to reject it for publication when issues related to its content below can be resolved. I cannot speak for (closed) investigations or social media which is not the content of this work, and we need to be wary of any potentially defamatory comments – unsubstantiated claims such as figure manipulation could be wrongly interpreted as such and would at best be removed or at minimum alternative explanations entertained.

2) [Because this matter has already been thoroughly investigated and adjudicated via a formal research ethics inquiry, we need to alert the editors of PeerJ to the fact that this manuscript submitted by During, Voeten, and Ahlberg, which repeats these allegations, is defamatory and potentially libelous] >>> It does not seem the case to me as only issues related to the original paper and its published data are made. The discrepancies highlighted and suggestions for recommended practices concerning them are discussed. I would ask the authors of this manuscript consider revising their manuscript such that it clearly acknowledges alternative interpretations of the issues raised, such as unintentional mistakes, database (copy/paste) errors, or graphing software misuse cannot be discounted. See also point 3.

3) [We also want the editors of PeerJ to be aware that this manuscript has been redundantly submitted to various journals by During and Ahlberg over the past two years and has already been published at least twice, …] >>>The work is available as a preprint which was reviewed and recommended by reviewers. Such work can be sent to a regular journal by choice of the authors. The journal, Academic Editor and reviewers were all aware of this and the authors also added a note about this in their submission). It is PeerJ policy that the original author be invited in case of replies; otherwise, this article could have been published like this. In addition, I feel feedback from the original author to these points would allow the inclusion of alternative interpretations / conclusions to be considered to the points raised to the fullest.

4) [The manuscript was also previously submitted to Scientific Reports which rejected it on the grounds that it provided no scientific value. The precedent set by the rejection, and its basis, should have bearing on the assessment of the suitability for publication of the same content that has now been submitted to PeerJ] >>> We have no public information on why Scientific Reports rejected the reply and there could be several and so this is hard to verify from our side and not mandatory. It is also not unusual for rejection of replies/rebuttals which could have various reasons. Irrespective, the original paper was published as it was and remains in the same way – the points raised here may therefore remain valid. I agree with the original PCI reviewer that it is unfortunate the issues raised were not picked up by reviewers, editors or mentioned by the authors, so a response paper remains valid if it can be demonstrated that they affect interpretations and revalidation of the results.

5) [Lack of disclosure was also problematic. It is clear that the authors concealed from PeerJ the true extent to which their manuscript had previously been submitted, breaching a policy of transparency that is typically standard and required by journals. Under the Declarations heading of the PeerJ submission, which asks “Are any elements of this paper or text under consideration at any other journal, or have they been published elsewhere already?”, During, Voeten, & Ahlberg concealed that they had previously submitted the same material to Nature Scientific Reports, as well as its submission in other online formats, including PubPeer. During, Voeten, & Ahlberg concealed multiple occasions on which they distributed the same material, falsely implying that the content had not already been exhaustively belabored and implying a greater novelty and applicability for the new submission. This misrepresentation to the journal, seemingly to make the manuscript appear more favorable for publication, is problematic.] >>> I am not familiar with the previous history of this manuscript at Scientic Reports and this is also not public knowledge. At PeerJ, the authors have listed all this information, and it was sent out for review, and it is therefore not appropriate to reject it based on this reason. PeerJ and I were fully aware of its history, and it fits with our guidelines and policy.

6) [In our experience, a critique would normally appear in the same scholarly journal as the paper that it critiques. The present attempt by During, Voeten, & Ahlberg to side-step that procedure and publish in a different journal is irregular, departs from standard practice, and should be regarded with concern] >>> I agree that it would be good practice that replies are published in the original venue, but this is often not the case (I am aware of multiple instances in PeerJ and various other journals). The reasons for this may be multifold and could for example, reflect a lack of, or conflict of interest of the journal among others.

7) [In the PeerJ manuscript draft, there is mention of the During et al., 2022 manuscript and its relation to the DePalma et al., 2021 paper being critiqued. However, almost none of the coauthors of During et al., 2022, except for During and Ahlberg, have joined the PeerJ critique draft or any of the aforementioned near-identical iterations that were circulated as complaints. Abandonment by nearly all other coauthors in those efforts strongly suggests their lack of confidence in the content being presented and the course of action. This should also be taken into consideration.] >>> This could reflect various things. It could also reflect a difference in opinion, expertise, priority, or conflict of interest. A conflict of interest could be the reason Jan Smith did not join as co-author of During et al. 2022 as well as co-authoring various publications with DePalma in 2019 and 2024. In other cases, it may reflect expertise and/or priority as various co-authors which did not join are experts in computed tomography and/or work in the synchrotron which is not the content of the points raised here. Without direct evidence from co-authors, this claim is hard to verify. Furthermore, there is no rule in our guidelines or principle that replies should involve all co-authors.

8) [Finally, the editor should be aware that two of the authors of this manuscript, During and Ahlberg, are currently under investigation at the University of Uppsala for alleged plagiarism of material that is closely related to this submission and additional serious breaches of professional ethics. During was the lead author on a manuscript that replicated the work that was reported by DePalma et al. 2021, that was already underway when During learned about it in 2017 (UoM investigation finding: “During was aware of DePalma’s long-standing research on seasonality and that he was working on a paper”, and “During was aware of DePalma’s work on seasonality and the use of isotope data prior to her visit”). Submission of the current manuscript to PeerJ is therefore at best a conflict of interest.] >>> As it is not public, this is hard to verify and so it cannot be used to influence this decision. If any of the allegations were to be substantiated, I would not support or condone them in any way. Irrespective of such investigations, allegations or investigations are not the content of the paper but rather the points raised from the original paper (if they remain valid) and suggestions on how they could be interpreted and dealt with in the future. Those points are at least considered valid by 2 external reviewers (1 re-reviewing from the pre-print) and the PCI recommender. I agree with Reviewer 1 that the rebuttal/response article furthers the conversation on the importance of data quality standards and good scientific conduct. Particularly if inconsistencies raised by Reviewer 2 can be addressed.

I feel multiple points are valid while other points need revision and additional clarification. If points raised are hard to substantiate, alternative interpretations should at least be entertained/discussed more fully or removed entirely. I would also recommend the authors to stick with only raising points related to the original paper which is currently the case (compare the reviewer 1). Please address all points raised by the reviewers.

Given the discrepancies between the reviews, please use my suggestions as a guideline:

[Basic reporting] The main points are clearly and concisely stated (compare reviewer 1). Some more supporting references should be given to support the recognition of seasonal growth patterns (compare reviewer 2: see further point).

[Experimental design] Not applicable or needed as it concerns a reply to a published paper using as much publicly available data as possible (compare reviewer 1).

[Validity of the findings] The authors disclosed the information about previous publication and peer-review through PCI and falls with the scope of the journal as it would otherwise not be sent out for review and considered as a reply if it was not.

[Stable isotope records with conflicting migratory signals] It should be more clearly stated that that points are not central to the core interpretation of the original paper (compare reviewer 2) but if they are stated in the original paper, they can be entertained in the rebuttal. Please add more clearly that you cannot rule out some points due to insufficient background data is available on migration or feeding habits of the animals in question or cite such information if available. Clearly state you assume similar principles in extinct fish feeding if this is the case and back it up with references. Add information which of conclusions of the original paper (e.g., Spring-Summer death?) would still stand if isotopic data is discarded.

[Primary Data] In my understanding the authors only state the original isotope data was not published with the original work. This is correct as the raw data (data points) of the original isotope study is not publicly available in published paper or currently available supplementary material. To avoid any misunderstanding, I would like you to add a qualifier (e.g., publicly available, or openly shared) to avoid any possible misunderstandings.

[Analytical Facility] Irrespective of the unfortunate nature of the lack of information on the analytical facility, it remains a fact that it is unknown and missing from the original article (compare reviewer 1). The known circumstances could be discussed more clearly but do not change these facts and one can ask the question if publication in lack of this information is appropriate. At least it hampers the full reproducibility of the original study.

[Methods] Irrespective of the unfortunate nature of the lack of detailed description of methods, it remains a fact that various details needed for reproducing them precisely are not known and missing from the article (compare reviewer 1). The known circumstances could be discussed more clearly but do not change these facts and one can ask the question if publication in lack of them is appropriate. At least it hampers the full reproducibility of the original study.

[Sampling Density and Amount of Carbon] I feel it should be acknowledged (ideally with supported reference(s)) that reliable analyses are within the limits for a skilled operator to recover the number of samples and well-run isotope lab to recover the data” but could also be mentioned that it is hard to reproduce when the operator or at least the lab is not known. The role of incremental step-size capabilities as opposed to burr/drill diameter on sampling should be discussed (see reviewer 2).

[Graphs in the Paper and Supplementary Materials] I feel the full possibilities of the reasons behind discrepancies between graphs in the original work should be acknowledged and discussed particularly those raised by reviewer 2. Please make sure the points raised and how they discussed are entirely consistent with the original publication when you refer them as such. Please rephrase when implying lack of integrity when this cannot fully be substantiated. One can wonder however why isotopic analyses need to be included in the original paper if its full details are not known and not needed at all to come to a similar conclusion. As this reply is focused on isotopic data, it would be good practice to at least point out that results and rougher estimate of time of death may hold if isotopic data is removed but would be more robust when confirmed by two different approaches.

[Thin sections in the Supplementary Materials] As manipulation seems hard to substantiate and alternative explanations exist, I recommend acknowledging this in the manuscript or remove this claim entirely. Something like – At first glance, image manipulation seems a possible explanation but alternative explanations such as … could explain this pattern (or cannot be ruled out entirely).

[Fish sizes] It is hard to verify these claims without fieldwork of my own but absence in such context may not necessarily be proof of absence. So, I feel you should consider if you really can raise this point (which seems to some degree a duplication of the known lack of raw data – line 216). If you want to raise it, you should more clearly highlight the potential limitations of available observations considering size and how those can be addressed in the future. This would include stating more clearly how many “published” fishes you are aware off from your own work and others is necessary to better understanding this point. Acknowledging the time, you spent in the field would be appropriate if you want to discuss this pattern and/or at least mentioning that an absence of smaller size in your observation could also related to having more limited sampling (make sure cited references are correctly cited in this content). Possibly you can also add information how confidently ontogenetic stages (juveniles versus adults) can be identified in this lineage.

[Conclusion of a Spring Death] Please explain in greater detail why you prefer the presence of osteocytes as opposed to the standard practice of counting lighter and darker growth bands. Additional supporting references should be cited in this context (see references suggested by reviewer 2 concerning standard practice of using growth bands in this lineage). Please make sure correct reference is made to the original study (Spring-Summer as opposed to Summer). As far as I understand your interpretations would narrow it down to a single season? If other observations (growth bands) in original paper led to similar conclusions, it makes me wonder why an isotopic analysis needed to be included at all in the original. However, both sclerochronological approaches as well as paleohistological approaches of fossil organisms are based on various assumptions. I feel the standard use, potential and limitations of both should be highlighted more clearly and back of by references.

[Conclusions] I feel that you can say that the available data and graphs are less conventional but not necessarily that they are not expected from direct analytical outputs (see comment by Reviewer 2). I suggest you rephrase this part to avoid misunderstanding. Line 249 (“Furthermore, indications of image manipulation are apparent in the supplementary material”) of the conclusion could be misunderstood as defamatory and difficult to demonstrate. The authors should note (line 206) that it may also have been an unintentional error – you do not present evidence to demonstrate that these sections have been flipped. Furthermore, it could have been mislabelled, miscopied into the Supplementary File, etc. Line 254 of the conclusion “We raised additional concerns regarding the authors’ failure to adequately discuss how their osteohistological slides, fish body size graphs, or stable isotope graphs support the conclusions of a spring death” needs to be reworded. Even if you consider the discussion inadequate, the most important point is whether the results support the conclusions, not whether the discussion supports the conclusions. I feel the absence of public raw isotopic data in the paper and supplementary materials can be highlighted (compare Reviewer 1) and can be discussed as a potential issue with reproducibility. It would be customary to mention that you obtained access to that data and its context if this is the case on demand. I suggest that the reply should focus on the isotopic data issues and embrace that other observations may give a range (spring-summer) but not necessarily a single season unless you can substantiate these points further with additional references and observations.

[References] Please make sure duplicate references are removed and add references where needed to support statements or interpretations (compare reviewer 2) more robustly.

Please address these points as well as all other points raised by the reviewers including those in annotated files.

Given the points raised above are beyond minor revisions, I consider it a case of major revisions and that the revised manuscript will be sent back into review.

I look forward to receiving your revised manuscript.

·

Basic reporting

In this manuscript by During et al., the authors document various irregularities, omissions, and implausible aspects of the results presented in a prior study by DePalma et al. which covered a very similar subject (and the same site). I reviewed an earlier pre-print version of this manuscript, and consequently many of the comments are going to carry-over from that, though I have noted a couple of additional minor comments (and have removed a number of comments I made in that initial review where the issue has since been addressed).

I think the authors use clear and unambiguous language throughout, their English is clear, literature references are provided, background details and context are given, and the information is relevant to the topics being discussed. As this is a rebuttal/response article, the figures are generally modifications or annotations of those from the DePalma et al. study, rather than original woks. Raw data is shared to the extent that it exists, and indeed one of the primary issues discussed is the lack of raw data in the study that this manuscript is rebutting, so that is also reflected here, but the authors here do got to lengths to provide data and literature support for claims that they are making.

Experimental design

As a rebuttal/response article, it does not have typical research questions and hypotheses, but is rather reviewing and critiquing issues in another study. Within that context though, the authors here do provide well defined questions/issues which they support by reviewing the results of the prior study and comparing them to other works in a rigorous manner. The way they performed their examinations of the DePalma et al results is provided in sufficient detail and should be broadly replicable.

Validity of the findings

Overall, I think the authors here make a compelling case that the results of DePalma et al.’s study are very irregular and largely irreproducible or verifiable at present. Their concerns are clear and concise. Indeed, it certainly raises questions concerning the editorial and peer-review process (not to mention formal rebuttal article process) of the journal which originally published the DePalma et al. study, given that the study lacks any reporting of its primary analytical data, has an unusually brief and vague Methods section, lacks any data availability statement, and of course also produced fairly unusual and in places implausible results plots (from which one would normally turn to the raw data to verify, but which one can of course not do in this case). As part of my review I provide some general and specific comments on the points raised by the authors and their broader implications, which I detail below.

I recommend During et al’s article for acceptance.

Additional comments

General Review / Comments:

I agree with the authors concerns regarding the notably incomplete methods section, which lacks information on sample weights, pretreatment and dissolution procedures, or analytical standards (all of which would be considered fairly basic reporting in a stable isotope geochemistry study; it is not uniquely unheard of for that information to omitted but it is unusual and generally a sign of poor practice for it to be missing).

I also agree that the amount of samples they obtained via microdrilling is unusually high given the small size of the specimens and the stated analytical process, and I agree that it too requires some corroboration, as such density of sampling would be difficult (to somewhat implausible) if done using ‘typical’ microsampling approaches and sample powder weights for d13C and d18O analysis of bioapatite structural carbonate on specimens of the size indicated. There may indeed be an explanation to that somewhere (and if so, I'd be very open to seeing it), but with the Methods almost entirely undescribed in the DePalma et al. study, I cannot immediately think of one that satisfies the concerns raised.

The above two issues flow naturally into the more glaring problem: namely that the primary isotopic data are not reported anywhere in the DePalma et al paper, nor are they available in their supplementary information.

In a general sense one cannot easily determine if a study is rigorous if no data are reported. While this would be more than enough concern on its own, it may be of greater concern in this instance given that it is my understanding that a university investigation (as reported in Dec 2023 in the news section in the journal Science) concluded that DePalma “drew the points in the graphs by hand” due him lacking access to the primary analytical data and instead working off of an “interim data sheet” which he only had in hard-copy. My understanding is that those actions were argued as necessary given that the collaborator running the isotopic analyses is now deceased and thus the actual raw data could not be accessed. I personally find that a bit odd, since every isotope lab I have worked with has, to my knowledge, kept fairly detailed digital records of their analytical runs, rather than exclusively keeping them on hard copy (or not at all), but perhaps that is not universal practice. Hand-drawing the results from printed hard-copies of data is an approach that is indeed sometimes used for digitizing old datasets, but for new analyses performed in the 2010s that seems unwarranted. As the authors of this response note, that process of seemingly manual plot generation appears to be a possible explanation for many of the issues observed in the results, such as the missing data points, the precision levels being inconsistent with both what is reported in the methods or predicted from the data points themselves, unusual stretching/distortion of some of the plots relative to others, and biologically/chemically unlikely levels of consistency across samples.

Basically, I agree with the authors here that the DePalma et al. results are sufficiently unusual/implausible that they cannot reasonably be taken at face value until primary data are presented to actually corroborate them (and ideally more details of exactly what methods were used and how the data were collected). Hopefully this isn't going too far beyond the scope of the specific issue, but the issues raised here unfortunately speaks to a perception that for various reasons appears present with much of the work (though certainly not all) done on the Tanis site: the combination of extraordinary claims, restricted access to material for outside verification, and frequently atypical analytical/data reporting that is often oddly lacking in the relative amount of primary data provided compared to more typical studies in the field (I say odd because given how extraordinary the material/site is said to be one would think that meeting or exceeding typical data reporting standards would be both easy and desirable). These sorts of issues have been discussed concerning reports of Tanis site geochemistry, stratigraphy/sedimentology, the fossil assemblage itself, and so forth. Indeed, further verification might go a long way to raising the perception/standard of research at this locality across the board, ideally including both confirmatory analysis and access by various outside research groups, such that there is no longer such a shadow hanging over the research done on the site as a whole. For the moment at least, I hope this rebuttal/response article furthers the conversation on the importance of data quality standards and good scientific conduct.

·

Basic reporting

Basic reporting was clear but misleading, as it circumvented known facts to arrive at unsupported conclusions. The authors lacked sufficient background in areas such as field work specifically related to this project and recognition of seasonal growth patterns in fish bone to adequately support their conclusions. The article was a belabored repeat of previous submissions that occurred after the issues raised in the draft were already adjudicated, and therefore does not constitute an example of professional scholarly work.

Experimental design

This manuscript did not involve scientific experimentation. The questions raised were not relevant and were instead misleading as they were not reflective of the full facts that were known by the authors. Because of that, and because the draft has already been published multiple times elsewhere, it does not fall within the aims and scope of the journal. The aforementioned belaboring of topics that have already been addressed, their redundant prior circulation in public domain, and failure of the authors to disclose those facts to PeerJ do not follow a pattern of altruistic impartial scientific pursuit and therefore do not follow a high ethical standard. The manuscript is a highly biased and uninformed attempt to recycle a re-review of a publication. The issues raised have been dealt with exhaustively elsewhere, and therefore it cannot truly be regarded as primary scientific research.

Validity of the findings

Problematically, the information presented by During, Voeten, and Ahlberg here is redundant, misleading, and repeats what has already been belabored. All sections exhibit insufficient support of the conclusions or deal with topics that have already been addressed. Comments below are organized by their respective sections.

[Stable isotope records with conflicting migratory signals]
Section summary comments

This section makes broad claims about the viability of interpreted migration signals in fossil organisms without acknowledging that insufficient background data is available on migration or feeding habits of the animals in question, or acknowledging that isotopic signals that may indicate migration, if reliable and reflective of said migration patterns, are not central to any key argument or core interpretation of the paper. This section fails to adequately support the authors’ criticism against the use of isotopic data to resolve seasonal cyclicity in the acipenseriform bones. Annual cyclicity is expected to be recorded in the bones of animals that experience annual fluctuations in any condition that affects bone growth, whatever those variables happen to be. For purposes of tracking annual cyclicity, identifying which variable affected the bone growth is of minimal importance compared to the fact that an annual pattern exists. During, Voeten, & Ahlberg point to questions related to oxygen isotope shifts that were tentatively interpreted to indicate migration patterns in the sturgeon, despite those interpretations having no bearing on the conclusions of the DePalma et al., 2021 study. Because not all sturgeon taxa exhibit migratory behavior, one would expect that a migration signal, if reliable and not an anomalous isotopic signature or artifact linked to other factors, would be inconsistently present for sturgeon among the population, thereby providing limited to no utility in supporting the conclusions of DePalma et al., 2021. For that reason, the interpretation of migration was never a key point in the study and was not relied upon for the conclusions. During, Voeten, & Ahlberg base part of their critique on patterns in carbon isotopes between the paddlefish and sturgeon in context of their dietary practices, which is problematic because there exists no detailed body of knowledge on extinct Mesozoic acipenseriform fish diets or feeding practices. Indeed, the specific environmental and metabolic factors that influence the annual fluctuations of bone growth even of extant acipenseriforms are very poorly understood (“…it is not easy to explore the effects of environmental and metabolic variations recorded in the spines because the processes governing bio-mineralization and growth of these pieces are still poorly known”; Meunier 2002; Bakhshalizadeh wt al., 2017). Because the authors lacked sufficient background knowledge on the feeding practices of the extinct fish, they attempted to rely on feeding data from some modern fish communities instead, without the ability to demonstrate that they can be applied to the fossil taxa. For example, many factors have never been demonstrated for fossil Mesozoic paddlefish or sturgeon, including whether they fed in the topwaters or bottomwaters, consumed live food or passively consumed detritus, possessed similar feeding practices, or feeding practices that differed markedly, etc. During, Voeten, & Ahlberg fail to adequately uphold their criticism that the isotope data from DePalma et al., 2021 does not support the conclusions that the fish perished in the Spring-Summer paleo equivalent. In addition, we furthermore point out that the conclusions of the DePalma et al., 2021 study are additionally in no way dependent on the isotopic data and the conclusions are fully supported even in its absence. (UoM investigation finding: “the overall conclusions of the paper still stand should the stable isotope results be removed”). The incorrect statements made by During, Voeten, & Ahlberg in this section, which fail to support their conclusions, demonstrate insufficient background/experience in this field of work by the authors.

[Primary Data]
Section summary comments
During, Voeten, & Ahlberg state that no isotope data is provided in the DePalma et al., 2021 paper, however they misrepresent themselves by their omission of certain keys facts. For example, as with many other studies, the data was available upon request and was, in fact, immediately supplied to the journal when requested. It was also supplied to the UoM during their rigorous investigation of the allegations made by During, Voeten, & Ahlberg. Not only did During, Voeten, & Ahlberg know about that, but they, too, received a copy of the data. The concealment of those three facts from PeerJ during submission is problematic and demonstrates a biased presentation of incomplete facts to support or imply an improper conclusion. While the data was always available upon request, the journal may have overlooked their opportunity to add a written mention of that fact to the paper during final edits, but it was never withheld. (UoM investigation conclusion: “’the low-resolution blurry photos of paper printouts’ provided by the Respondents to Scientific Reports and the Panel of Investigation…were consistent with what would be expected as a summary of calibrated data reported back to a client from a service isotope lab”). These data sheets nonetheless constituted scientific data by any definition. (Additional UoM conclusion: “there was independent evidence from 5 individuals (Oleinik, Burnham, Cichocki, Larson, Erikson) that DePalma’s isotope data pre-dated During’s visit to Tanis in 2017 and that one (Oleinik) had seen the plots in 2016 or early 2017 and confirmed they were the same ones that appear in the Scientific Reports paper”.

[Analytical Facility]
Section summary comments

During, Voeten, & Ahlberg state that some details related to the contributions by our late colleague are unclear due to his untimely decease. While regrettable, this, too, was previously surmounted in the UoM investigation, which was known by During, Voeten, & Ahlberg but not disclosed by them. Our late colleague facilitated the completion of various specialized tasks that were closer to his area of expertise than anyone else on the paper at that time. (UoM investigation conclusion: “although McKinney’s institution did not have the kind of apparatus supposedly used for the analysis, this was not evidence that McKinney had not sent the samples elsewhere for analysis”, in addition to the assessment that “’the low-resolution blurry photos of paper printouts’ provided by the Respondents to Scientific Reports and the Panel of Investigation…were consistent with what would be expected as a summary of calibrated data reported back to a client from a service isotope lab”.

[Methods]
Section summary comments

As mentioned regarding the analytical facility, the death of our colleague Curtis Mckinney occurred slightly before full completion of his contribution, which would have included his write-up of the isotopic methods. Using the notes and discussions that we had during the process, the methods were reconstructed as fully as possible, but some portions remained unknown. Our attempt was to compile methods that were as complete as those notes and discussions would allow, while not including any known falsehoods. This, too, was raised and surmounted in the UoM investigation, the process and outcomes of which are known to During, Voeten, & Ahlberg and which they failed to disclose.

[Sampling Density and Amount of Carbon]
Section summary comments

During, Voeten, & Ahlberg here fail to support their proposed conclusion that the sampling density is incompatible with the ability to retrieve sufficient sample for analysis. What they concealed from this section is that they are aware of multiple additional isotopic experts who were consulted and determined that it would in fact be possible to functionally retrieve sufficient sample. The description of drill bit shapes and sizes demonstrates an unfamiliarity with the process and technique, as a progressive inward-directed sampling along a peripheral transect is limited only by the incremental step-size capabilities of the micromill, and not the diameter of the burr. It is therefore completely unclear how exactly the number of samples per length of the sampling transect can “correspond”, as claimed by During at al. to the drill diameter. No such correlation exists. This, too, was raised and surmounted in the UoM investigation, the process and outcomes of which are known to During, Voeten, & Ahlberg and which they failed to disclose. The UoM investigation conclusion (which During, Voeten, & Ahlberg are aware of) further contradicted the claims by those authors in the present draft, establishing that “typically mass spectrometers require 25ug of carbonate for reliable analysis (approximately 500ug of fresh bone is required to reliably yield >20ug carbonate). However, it is just within the bounds of possibility for a skilled operator to recover the number of samples, and for a well-run isotope lab to recover the data”.

[Graphs in the Paper and Supplementary Materials]
Section summary comments

During, Voeten, & Ahlberg point to a variety of errata exhibited by the graphed isotopic data, however, more troublingly, they made no mention that every point raised in this section was laboriously addressed, discussed, explained, and adjudicated in the UoM investigation, all the details of which are known by During, Voeten, & Ahlberg. Their conscious and willful concealment misrepresents the facts of the issue and creates an intentional bias that otherwise could have been avoided. While the topics in this section have already been fully and thoroughly dealt with, it is worthwhile to point out some facts in response. For example, there do indeed exist a number of errata that ultimately result primarily from the untimely death of our colleague and the effort to organize his work, including the manual transcription of figures from his data sheets. The errata are largely applicable to graphs, graphed points, etc., that were manually transcribed from his printed data sheets as carefully as possible. (UoM investigation conclusion: “The inconsistencies in the data were explained as genuine errors resulting from the lack of raw data as a consequence of the death of McKinney and DePalma’s use of the interim data sheet to hand-draw the graphs”. Other criticisms noted in this section deal with factors that the authors (During, Voeten, & Ahlberg) evidently did not understand or fully read in DePalma et al., 2021 or the UoM investigation report. For example, as explained previously in the UoM investigation and known already to During, Voeten & Ahlberg, none of the graphs are identical. Because the specimens came from multiple animals with near identical life histories from a synchronous death assemblage, and in some instances multiple sets from the same individual, similarities in patterns are not only perfectly normal, but they are expected. Because the growth lines in fish bones are wavy and sinuous, they exhibit compressed or expanded representations of the same growth band pattern, for example in troughs as opposed to lobes. So, in one individual, a single sliced surface can contain a 5-band pattern of growth that is perhaps 3 mm thick in one expanded region or 0.3 mm thick in an adjacent compressed region, with every variation in between. Another example that During, Voeten, & Ahlberg cite is the use of data point icons in the supplemental materials that they misinterpreted as error bars because they did not read the actual figure caption that explicitly states so. Another example is the mention of one specimen number that is listed twice, which was previously clearly explained, and demonstrated during the UoM investigation as being correct, because it reflected multiple sampling of the same specimen, however During, Voeten, & Ahlberg concealed that fact, implying a discrepancy that pointed toward lack of integrity. (UoM investigation comment: “The Appeal Panel requested clarification from DePalma who explained that Fig 2 was a repeat analysis of the same sample, and provided the data from which it was plotted”. At the time of their submission to PeerJ, During, Voeten, & Ahlberg knew this, and every single other detail in their section about the graphs, which they could have transparently revealed to the journal but instead they only mentioned their initial criticism and omitted everything that transpired since then in the process of addressing and/or satisfying it. As we had previously mentioned during the UoM investigation, of which During, Voeten, & Ahlberg are aware, the legitimate errata in the graphs that are linked to the manual transcription of the data resulted in near imperceptible shifts that were not sufficient to affect the conclusions of the study in any way whatsoever. (UoM investigation conclusion: “the differences between curves derived from the numerical values in the tables and the curves published in the Scientific Reports paper were minor, as confirmed by two independent individuals (Oleinik and Smit – During’s MSc advisor) and were likely the result of the published curves being copied from original plots”. The UoM investigation upheld the assessment that data fabrication/manipulation had not occurred, even after an appeal process initiated by During & Ahlberg, and went on to comment that “the overall conclusions of the paper still stand should the stable isotope results be removed”.

[Thin sections in the Supplementary Materials]
Section summary comments

In this section, During, Voeten, & Ahlberg claim that DePalma et al manipulated an image in the supplementary section, flipping an image that had been photographed twice from the same side. Not only is this statement false and insufficiently supported, but the UoM investigation thoroughly looked into it with multiple experts, concluding that manipulation had not taken place, that the images were not both of the same face of the slide, and that the claim was unfounded. Furthermore, During was cautioned by the UoM Chair of Ethics, that, in light of During, Voeten, & Ahlberg being aware that their claim was false, if they were to perpetuate that claim they would be knowingly and intentionally circulating false and misrepresented facts in a malicious way.

[Fish sizes]
Section summary comments

During, Voeten, & Ahlberg claim that During observed no juvenile or sub-yearling fish while she was on-site, implying that none were there. This statement, at best, indicates that they did not observe or recall the sub-yearling acipenseriform fish skull included in Figure 1 of DePalma et al., 2021 as a Micro-XRF map. At worst, the statement reflects a willing concealment of their knowledge of sub-yearling fish fossils at the site, including the personal experience of During. In addition, if During did not observe any sub-yearling fish during her brief site visit, that does not indicate that the fish were not present, but rather is reflective of During’s inexperience working or identifying fossils in that field setting. During lacks familiarization or experience with field work in the Hell Creek Formation (we were told that her ~10-day site visit in 2017 was her first experience in the Hell Creek Formation and she had accrued fewer than two week’s additional field time there in the 7 years since then, as far as we are aware). As a result, it was a challenge for During to detect most fossil material and it had to be shown to her on many occasions. She even personally encountered sub-yearling fish herself, although not intentionally- the well-preserved fins and partial body of what would have been a ~14 cm juvenile acipenseriform fish were discovered freshly broken in her debris pile, unintentionally destroyed as the fossil went unnoticed while she dug into the outcrop [FIG 1]. While During may not have measured any fish lengths during her short visit, other researchers on-site did. During, Voeten, & Ahlberg claim that the smallest fish at the site are all 15 cm long, and in support of that they erroneously cited a graph from another publication that intentionally began its tally at the 15 cm size range because the densely tangled mass-death assemblage made it problematic to accurately tally fish smaller than that in situ prior to preparation of the blocks. Regarding growth estimates, the comparison with the seasonal growth ranges of modern fish involved clearly citing multiple compiled ichthyological works, contrary to claims by During, Voeten, & Ahlberg.

[Conclusion of a Spring Death]
Section summary comments

In this section, During, Voeten, & Ahlberg state that they did not clearly observe osteocytes in the fish bone images, and therefore conclude that discerning the season of death would be challenging or impossible. This claim is perplexing, clearly demonstrates their unfamiliarity with the subject material, and demonstrates their inability to sufficiently support their claim. The assessment of seasonal oscillation in the bones of fish (including acipenseriforms) is routinely, reliably, and most-frequently carried out via counting the annuli, or growth bands, in the bone cross-sections (Kolhorst et al., 1980; Brennan & Cailliet, 1989; Jackson et al., 2007; Bakhshalizadeh et al., 2011; Neely & Lynott, 2016; FIG 2A). Each year is represented by a couplet comprised of a light and dark band, with the denser, dark bands grown during the favorable growth period (Spring-Summer) and the lighter, more-translucent, less-dense bands grown during the unfavorable growth period (Fall-Winter). For During, Voeten, & Ahlberg to suggest that the seasonal assessment should be based on osteocyte distribution is peculiar, uncustomary, not requisite, and not aligned with standard practice. Our assessment, based on the very clear patterns exhibited by strongly defined growth band couplets that are on par with modern counterparts [FIG 2] is robust, unambiguous, and was favorably critiqued by multiple professional bone histologists prior to publication of DePalma et al., 2021 (and listed in the acknowledgements thereof). Based on studies of annual growth in modern acipenseriformes, the bone histological data, alone, was sufficient as the exclusive indicator of season-of-death for DePalma et al., 2021, barring all other evidence presented by that study. In addition, During, Voeten, & Ahlberg attempt to undermine the conclusions of DePalma et al., 2021 by claiming that the data indicates a Summer death rather than Spring. While we disagree with the focused exclusivity of that assessment, their comment further indicates their unfamiliarity with the DePalma et al., 2021 paper, which interpreted a Spring-Summer range for time of death, which is consistent both with the data of DePalma et al., 2021 and During’s replication of the process.

[Conclusions]
Section summary comments

During, Voeten, & Ahlberg state that “The stable isotope graphs presented in DePalma et al.'s paper, as depicted in Figure 25 and the Supplementary Materials, exhibit patterns that deviate from what would be expected from direct analytical outputs”. With that statement, During, Voeten & Ahlberg misrepresent themselves by concealing that they already knew the reasons for minor errata in the graphs, as they had already been thoroughly addressed during the UoM investigation. They also cite absence of raw data, when they already had in their hands the raw data, which was immediately provided upon request, as is customary for any scientific paper that does not contain it in the text or supplement. They claim that the evidence presented does not align with the assertion/conclusions of our study, however that statement is incorrect and fully unsupported, because the multiple, independent, mutually reinforcing lines of evidence in DePalma et al., 2021 robustly support the conclusions of a Spring-Summer time of death. During, Voeten, & Ahlberg claim that the osteohistological slides, fish growth, and isotope data fail to indicate the season of death, however those are all thoroughly discussed and explained in the DePalma et al., 2021 paper, including graphical explanations of the histological subdivisions laid down during favorable vs lean growth periods and showing that the fish died at some point during the favorable period (Spring-Summer). That assessment was also made by the UoM investigation, which found that “the overall conclusions of the paper still stand” even in absence of the isotopic data. system for personal reasons. As such, the manuscript of During, Voeten, & Ahlberg is not fit for publication and would not serve any useful benefit if that were to be published. Regardless, it can already be viewed via multiple other hyperlinks, removing any justification for adding yet another.

[References]
The Reference portion of the manuscript submitted to PeerJ by During et al, with only 31 items in it, contains an error indicative of unprofessional editing of the manuscript. References #20 (Lines 320-323) and #22 (Lines 329 – 332) are identical and simply redundant

Closing remarks
We reiterate that During, Voeten, & Ahlberg were fully aware of the answers of everything raised in the manuscript draft that they submitted to PeerJ, that the points have been redundantly belabored time and again over various online formats, we note that During, Voeten, & Ahlberg claim that those points have already have been peer reviewed, and the contents of this manuscript have been previously addressed during a thorough, laborious, and robust year-long UoM investigation. As such the submission of this manuscript to PeerJ deviates from the pattern of altruistic scholarly duties and more closely aligns with a pattern of abuse of the journal system. This statements made by During, Voeten, & Ahlberg in their manuscript draft were presented to the journal in such a way as to be knowingly misleading because the authors had full prior knowledge that the information they presented was obsolete. They knowingly omitted any clarifications that would surely have altered the viability of their conclusions, and furthermore During, Voeten, & Ahlberg are fully aware of the official outcome of the UoM investigation, which addressed the claims that they here repeat. During, Voeten, & Ahlberg circumvented transparency by not disclosing their prior knowledge of the UoM investigation outcome when they submitted their draft to PeerJ because otherwise their critique would have been undermined. Omission of those facts goes beyond an honest lack in rigor and instead is a conscious concealment of key facts that would weaken or invalidate their argument. Every criticism except for comments on the size of fish had been previously incorporated into the UoM investigation that was prompted by During & Ahlberg and they have all been adjudicated. During, Voeten, and Ahlberg present their old complaints here, as if they are fresh ideas, knowing that they and their basis are now obsolete. The data provided by the authors in support of their criticisms is incomplete and does not reflect their full knowledge of the situation’s state of development. The study lacks both impact and novelty and instead is a repetition of prior-stated arguments that have already been investigated and addressed. There is no meaningful replication of any prior work or concepts and this study does not provide any meaningful advance. We recommend that this manuscript and its contents are not fit for publication.
We have followed the proper procedure thus far in cooperation with our colleagues and the UoM investigation, we were found innocent of all allegations, including the ones that comprise the body of this manuscript draft, the lead author of DePalma et al., 2021, Robert DePalma, has already complied with the UoM disciplinary recommendations related to several instances of poor research practice related to challenges in properly handling the data of our late colleague (assessments that were not part of the 4 allegations by During & Ahlberg), and we continue to follow the proper procedure by coordinating with the journal to take any necessary clarifying steps. The manuscript submitted by During, Voeten, & Ahlberg, the contents of which have already been publicly circulated multiple times in multiple ways, serves no purpose in being published, represents an obsolete moment in the timeline, and would in fact potentially interrupt the process of completing our requisite tasks in conjunction with the journal.
Figure 1. A & B, a partial well-preserved body wall and fins from a sub-yearling acipenseriform fish with fresh breaks at the matrix edges, discovered in the debris pile of Melanie During in August 2017 during her ~10-day visit to the site. The fish, before unintentional destruction during excavation, would have been approximately 13-15 cm in length based on comparative body proportions (bottom).

Figure 2. Standard methodology to establish multi-year chronology in modern fish relies on counting the opaque and translucent growth band couplets (annuli) as shown in the thin section of a modern fish (A). The growth bands in the fossil specimens are equally well-resolved and robust (B), enabling equivalent interpretation. A, from Brennan & Cailliet, 1989; B, from DePalma et al., 2021, sup mat figure 5. Image in (B) converted to greyscale for ease in comparison with (A).

References

Bakhshalizadeh, S., Bani, A., Abdolmalaki, S., Nahrevar, R., & Rastin, R. (2011). Age, growth and mortality of the Persian Sturgeon, Acipenser persicus, in the Iranian waters of the Caspian Sea.
Bakhshalizadeh, S., Bani, A., Abdolmalaki, S., & Moltschaniwskyj, N. (2017). Identifying major events in two sturgeons’ life using pectoral fin spine ring structure: exploring the use of a non-destructive method. Environmental Science and Pollution Research, 24, 18554-18562.
Brennan, James S., and Gregor M. Cailliet. "Comparative age-determination techniques for white sturgeon in California." Transactions of the American Fisheries Society 118, no. 3 (1989): 296-310.
DePalma R et al. (2021). Seasonal calibration of the end-cretaceous Chicxulub impact event. Scientific reports, 11(1), 1-9.
During M. et al. (2022) The Mesozoic terminated in boreal spring. Nature, 603(7899), 91-94.
Jackson, N. D., Garvey, J. E., and Colombo, R. E. (2007). Comparing aging precision of calcified structures in shovelnose sturgeon. J. Appl. Ichthyol. 23, 525–528. doi: 10.1111/j.1439-0426.2007.00875.x
Kohlhorst, D., Miller, L., and Orsi, J. (1980). Age and growth of white sturgeon collected in the Sacramento-San Joaquin Estuary, California: 1965-1970 and 1973-1976. Calif. Fish Game 66, 83–95.
Meunier FJ (2002) Skeleton. In: Panfili J, de Pontual H, Troadec H, Wright PJ (eds) Manual of fish sclerochronology. Ifremer-IRD Coedition, Brest, pp 65–88
Neely, B. C., & Lynott, S. T. (2016). Examination of the world record flathead catfish captured from Elk City Reservoir, Kansas, in May, 1998. Transactions of the Kansas Academy of Science, 119(3-4), 353-359

Additional comments

Review of PeerJ manuscript 98788 entitled “Calibrations without raw data- a response to “seasonal calibration of the end-Cretaceous Chicxulub impact event”, authored by M. During, D. Voeten, & P. Ahlberg, that critiques the work of Nature Scientific Reports manuscript by DePalma et al., 2021 entitled “Seasonal calibration of the end-Cretaceous Chicxulub impact event”.

1) The issues raised in this manuscript are a repetition of allegations raised in a complaint filed by During and Ahlberg at the University of Manchester on December 9, 2022. An exhaustive investigation was carried out by the University of Manchester which included an external specialist and an appeal process. The University of Manchester, after considering extensive documentation and live witness testimony, produced a comprehensive report dealing with those allegations. The pertinent results contained in that report were published online on December 15, 2023, please see: https://www.manchester.ac.uk/discover/news/palaeontologist-cleared-of-fabricating-data-in-dino-killing-asteroid-paper/

The mistakes made in graph preparation and reasons for the truncated methods section are all clearly explained in that document and therefore this submission to PeerJ provides nothing of scientific value to the reader beyond what is explained in that report which has now been available online for 4 months.

For reasons of transparency, we are obligated to list all of the allegations that were presented by During & Ahlberg, which comprised the UoM investigation:

-Scooping and unethical sabotage
During and Ahlberg accused Robert DePalma of “deliberately publishing a rival paper on the same topic and identical conclusions” as an “unethical sabotage attempt” against Melanie During.

-Data fabrication and manipulation
During and Ahlberg accused Robert DePalma of data fabrication and manipulation in order to carry out the “scoop and unethical sabotage” of the first allegation

-Corrupt interaction with handling editor
During and Ahlberg accused Robert DePalma of corrupt interaction with the journal in an effort to carry out the action of the first allegation

-Smear campaign
During and Ahlberg accused Robert DePalma of initiating a social media smear campaign against them

After the official UoM inquiry that was exceptionally rigorous and lasted nearly one year, plus a subsequent appeal initiated by the complainants and a thorough treatment of every single allegation, all four of the allegations were found to be without merit and were overturned.

The investigation concluded that the data in question were not fabricated, the claims of “scooping via a rival paper”, corrupt interactions with the journal, and social smear campaign were all unsupported by the evidence.

The investigative panel determined that, even in light of this exoneration, “the overall conclusions of the paper still stand should the stable isotope results be removed”

It was determined by the investigation that During also knowingly provided untrue statements to the panel, including but not limited to (1) claiming that she was not aware that DePalma was working on the study, and (2) claiming that he had not invited her to be a part of it.

After all of the points listed against the defendants were addressed and/or overturned, the UoM Ethics Board clearly informed During and Ahlberg that, given that they were aware that their claims were overturned by the outcome ruling of the investigation, they were to refrain from pressing them further as that would constitute dishonest and malicious willful dissemination of untrue statements.

2) Because this matter has already been thoroughly investigated and adjudicated via a formal research ethics inquiry, we need to alert the editors of PeerJ to the fact that this manuscript submitted by During, Voeten, and Ahlberg, which repeats these allegations, is defamatory and potentially libelous. In light of the severity and nature of the allegations that were leveled at the corresponding authors of DePalma et al., 2021 by During and Ahlberg in their complaint to UoM, and the fact that all of those allegations were overturned and found without merit, the additional attempts by During and Ahlberg to repeatedly circulate the same critique, including the draft submitted to PeerJ, are unethical, fail to honor the official process that had been undertaken, and do not follow a pattern of impartial and altruistic academic pursuit. They instead appear to constitute an abuse of the scholarly journal system for reasons other than scientific merit.

3) We also want the editors of PeerJ to be aware that this manuscript has been redundantly submitted to various journals by During and Ahlberg over the past two years and has already been published at least twice, see for example:

PubPeer (https://www.pubpeer.com/publications/9B9D041BD4D3633C2D4F99D002DF87),

PCI Paleo (DOI:10.24072/pci.paleo.100221)

These two versions of this manuscript are essentially the same as what has been submitted to
PeerJ, in fact the PCI Paleo manuscript is verbatim the same and was published online on the 26th of March, 2024, only ten days before the date of April 5th, 2024 assigned to the PeerJ submission.

Therefore the authors are attempting to re-publish the same material which has already been published and has an active DOI assigned. This does not meet the criterion for valid new research that would be suitable for publication.

4) The manuscript was also previously submitted to Scientific Reports which rejected it on the grounds that it provided no scientific value. The precedent set by the rejection, and its basis, should have bearing on the assessment of the suitability for publication of the same content that has now been submitted to PeerJ.

5) Lack of disclosure was also problematic. It is clear that the authors concealed from PeerJ the true extent to which their manuscript had previously been submitted, breaching a policy of transparency that is typically standard and required by journals. Under the Declarations heading of the PeerJ submission, which asks “Are any elements of this paper or text under consideration at any other journal, or have they been published elsewhere already?”, During, Voeten, & Ahlberg concealed that they had previously submitted the same material to Nature Scientific Reports, as well as its submission in other online formats, including PubPeer. During, Voeten, & Ahlberg concealed multiple occasions on which they distributed the same material, falsely implying that the content had not already been exhaustively belabored, and implying a greater novelty and applicability for the new submission. This misrepresentation to the journal, seemingly to make the manuscript appear more favorable for publication, is problematic.

6) In our experience, a critique would normally appear in the same scholarly journal as the paper that it critiques. The present attempt by During, Voeten, & Ahlberg to side-step that procedure and publish in a different journal is irregular, departs from standard practice, and should be regarded with concern.

7) In the PeerJ manuscript draft, there is mention of the During et al., 2022 manuscript and its relation to the DePalma et al., 2021 paper being critiqued. However, almost none of the coauthors of During et al., 2022, except for During and Ahlberg, have joined the PeerJ critique draft or any of the aforementioned near-identical iterations that were circulated as complaints. Abandonment by nearly all other coauthors in those efforts strongly suggests their lack of confidence in the content being presented and the course of action. This should also be taken into consideration.

8) Finally, the editor should be aware that two of the authors of this manuscript, During and Ahlberg, are currently under investigation at the University of Uppsala for alleged plagiarism of material that is closely related to this submission and additional serious breaches of professional ethics. During was the lead author on a manuscript that replicated the work that was reported by DePalma et al. 2021, that was already underway when During learned about it in 2017 (UoM investigation finding: “During was aware of DePalma’s long-standing research on seasonality and that he was working on a paper”, and “During was aware of DePalma’s work on seasonality and the use of isotope data prior to her visit”). Submission of the current manuscript to PeerJ is therefore at best a conflict of interest.

For the eight above reasons, we recommend that the submitted manuscript is not a fit candidate for publication. However, that assessment is supported by multiple additional key facts related to the content of the manuscript. Below, following the format for PeerJ review, we address specific examples of concerns with the During, Voeten, & Ahlberg manuscript draft, and address their inappropriate use or otherwise unsuitability for publication.

---

## Round 0.2 · Minor Revisions

The authors have thoroughly addressed nearly all of the extensive comments raised by the prior Editor and the previous round of review, which comprised conflicting assessments of this manuscript (reject decision, accept decision). Reviewer 2 has provided further comments, on this revised version, that raise concerns over the authors' previous responses to their concerns.

On review of the re-submitted manuscript, focusing closely on the recommendations made by the Editor, I consider that several points remain in need of attention prior to acceptance. I encourage the authors to please address those (provided in my attached document).

·

Basic reporting

Basic reporting was clear but misleading, as it circumvented known facts to arrive at unsupported conclusions. The authors lacked sufficient background in areas such as field work specifically related to this project and recognition of seasonal growth patterns in fish bone to adequately support their conclusions. The article was a belabored repeat of previous submissions that occurred after the issues raised in the draft were already adjudicated, and therefore does not constitute an example of professional scholarly work. The revised manuscript did not adequately address the issues raised by the first review.

Experimental design

This manuscript did not involve scientific experimentation. The questions raised were not relevant and were instead misleading as they were not reflective of the full facts that were known by the authors. Because of that, and because the draft has already been published multiple times elsewhere, it does not fall within the aims and scope of the journal. The aforementioned belaboring of topics that have already been addressed, their redundant prior circulation in public domain, and failure of the authors to disclose those facts do not follow a pattern of altruistic impartial scientific pursuit and therefore do not follow a high ethical standard. The manuscript is a highly biased and uninformed attempt to re-review a publication. The issues raised have been dealt with exhaustively elsewhere, and therefore it cannot truly be regarded as primary scientific research.

Validity of the findings

Problematically, the information presented by During, Voeten, and Ahlberg here is redundant, misleading, and repeats what has already been belabored. All sections exhibit insufficient support of the conclusions or deal with topics that have already been addressed. This has not been sufficiently remedied since the first review, and the resubmitted draft has failed to address issues or inadequacies raised by the first review and Editor. Comments below are organized by their respective sections.

Additional comments

Dear Editor,
Thank you for inviting us to review the revised version of this manuscript. I appreciated the opportunity to provide a first round of review for the first draft of this manuscript. We note that the revised version of the manuscript remains problematic in many ways. The editors recommended major revisions after the first round of review, with which I concurred based on many issues, inconsistencies, poorly supported or unsupported claims, and misdirection that I had outlined in the first review. After reading the revised draft and commentary by the authors, it is unfortunately clear that the required major revisions were not carried out.
Following the opening comments below is a review of the text in PeerJ format, and a closing statement. Please be aware I have consulted my PhD supervisors at the University of Manchester, Professors Phil Manning and Roy Wogelius, as well as my colleague at Florida Atlantic University, Professor Anton Oleinik, in the course of preparing this review. Many critiques and viewpoints shared in the first review remain valid and are carried over here.

Sincerely,
Robert A. DePalma
>>>>>>>>>>>>>>>>>>>>>>>>>>>>>>>>>>>>>>>>>>>>>>>>>>>>>>>>>>>>>>>>>>>>>>>>>>>>>>>>>>>>>

Review of PeerJ manuscript 98788 entitled “Calibrations without raw data- a response to “seasonal calibration of the end-Cretaceous Chicxulub impact event”, authored by M. During, D. Voeten, & P. Ahlberg, that critiques the work of Nature Scientific Reports manuscript by DePalma et al., 2021 entitled “Seasonal calibration of the end-Cretaceous Chicxulub impact event”.
In the revised draft, the authors have failed to adequately or fully address the recommendations and concerns posed by the journal and the reviewers. In many instances, a superficial effort was made to placate editor requirements and reviewer comments but those efforts fell short of the editorial requirement of “major revisions”, and were unsatisfactory to remedy the issues raised during the first review. This included an instance in which the authors changed only a single word, providing a cursory and superficial action that failed to markedly change the text or remedy the insufficiencies in their work that prompted the editorial and reviewer requests. In other instances, the authors directly contradicted the editors or declined to address or remedy the issues that were raised by the editor. Untrue or poorly supported comments were also provided by the authors. In yet other instances, the authors demonstrated further unfamiliarity with standard techniques or insights, which carried over from similar unfamiliarities that were made apparent in their first draft, including unfamiliarity with standard methods of histologically determining the season of death in fish. In total, the authors failed to improve their paper to a point that sufficiently satisfied the critiques and recommendations of the reviewers as well as editorial requests, such that this paper is not fit for publication. of This reviewer strongly recommends that, based on those reasons alone, this draft is not fit for publication.
There are, however, other concerns regarding the submitted manuscript. This draft was submitted by the authors under misleading and dishonest pretenses, in that the authors submitted it while in possession of pertinent information that would serve to invalidate the relevance and utility of their manuscript.
Specifically, the authors were aware of the following key details (and others) regarding the outcome of a year-long rigorous investigation by the University of Manchester, regarding the same research that constitutes the topic of their PeerJ draft, which determined that:

1) The data in the DePalma et al., 2021 study was genuine and not fabricated
2) Minor errata existed
3) The minor errata did not impact the conclusions of the study in any way
4) A course of action was proscribed to correct the errata and clarify any questions, via coordinated arrangements between the University of Manchester, the authors, and the journal Scientific Reports.
5) The DePalma et al., 2021 authors have been working to submit an erratum to the journal.

The authors (During, Ahlberg, and Voeten) are and have been aware of all of those facts, and that the authors of DePalma et al., 2021 are in the process of dutifully following the scientific method and the course proscribed by UoM and the journal, working in tandem to offer full clarification, correction of minor errata where necessary, and additional data that incontrovertibly demonstrates the veracity of the original work.

Despite knowing that, they (During, Ahlberg, and Voeten) still sought to submit the present manuscript. The submission therefore does not follow a pattern that seeks to collegiately correct the scientific record or address questions that they raised, since they are aware that measures are already being taken to address any minor errata and clarify any misconceptions.

Furthermore, the present draft is also a near-identical re-hash of multiple grievance documents submitted elsewhere by the same authors that all redundantly and repetitively belabor the same points time and again, while they were fully aware that the matters being discussed were either demonstrated false or that corrections were in the process of being made to minor errata via the procedures proscribed by the UoM and Scientific Reports. The authors dishonestly conceal knowledge of those facts and proceed as if the document that they submitted holds merit, when in fact it does not serve any useful purpose.

The authors of the draft were early-on made aware, prior to making their first 2022 public complaint about aspects of the DePalma et al., 2021 study, that the DePalma et al., 2021 team would address any legitimate minor errata, that the scientific method would be followed to responsibly and properly carry out those efforts, and that they (During, Ahlberg, and Voeten) were even invited by DePalma et al., 2021 to join together in that corrective effort. Instead, the authors (During, Ahlberg, and Voeten) declined to follow the constructive, scientific, and collegiate path and elected instead to choose an unconstructive path of public denouncement that misleadingly withheld their full knowledge of the situation, and while knowing that doing so would, without question, serve to delay and hamper the already-begun effort of DePalma et al. to properly and scientifically correct any minor errata in the proper fashion. Their choice to repeat it in public forum, time and again, each time hampering that process, therefore does not follow a pattern of seeking out a correction of the scientific record, but knowingly serves to slow it, demonstrating that their reasons for doing so must be something other than altruism for the sake of science. This draft is the latest iteration in that pattern.

Below please find additional comments regarding the revised manuscript, which are much in-line with the first review, as very little has been improved.

1) The issues raised in this manuscript are a repetition of allegations raised in a complaint filed by During and Ahlberg at the University of Manchester on December 9, 2022. An exhaustive investigation was carried out by the University of Manchester which included an external specialist and an appeal process. The University of Manchester, after considering extensive documentation and live witness testimony, produced a comprehensive report dealing with those allegations. The pertinent results contained in that report were published online on December 15, 2023, please see: https://www.manchester.ac.uk/discover/news/palaeontologist-cleared-of-fabricating-data-in-dino-killing-asteroid-paper/

The minor errata made in graph preparation and reasons for the truncated methods section are all clearly explained in that document and therefore this submission to PeerJ provides nothing of scientific value to the reader beyond what is explained in that report which has now been available online for 9 months.

For reasons of transparency, we are obligated to list all of the allegations that were presented by During & Ahlberg, which comprised the UoM investigation:

-Scooping and unethical sabotage
During and Ahlberg accused Robert DePalma of “deliberately publishing a rival paper on the same topic and identical conclusions” as an “unethical sabotage attempt” against Melanie During.

-Data fabrication and manipulation
During and Ahlberg accused Robert DePalma of data fabrication and manipulation in order to carry out the “scoop and unethical sabotage” of the first allegation.

-Corrupt interaction with handling editor
During and Ahlberg accused Robert DePalma of corrupt interaction with the journal in an effort to carry out the action of the first allegation

-Smear campaign
During and Ahlberg accused Robert DePalma of initiating a social media smear campaign against them

After the official UoM inquiry that was exceptionally rigorous and lasted nearly one year, plus a subsequent appeal initiated by the complainants and a thorough treatment of every single allegation, all four of the allegations were found to be without merit and were overturned.

The investigation concluded that the data in question were not fabricated, the claims of “scooping via a rival paper”, corrupt interactions with the journal, and social smear campaign were all unsupported by the evidence.

The investigative panel determined that, even in light of this exoneration, “the overall conclusions of the paper still stand should the stable isotope results be removed”

It was determined by the investigation that During also knowingly provided untrue statements to the panel, including but not limited to (1) claiming that she was not aware that DePalma was working on the study, and (2) claiming that he had not invited her to be a part of it.

After all of the points listed against the defendants were addressed and/or overturned, the UoM Ethics Board clearly informed During and Ahlberg that, given that they were aware that their claims were overturned by the outcome ruling of the investigation, they were to refrain from pressing them further as that would constitute dishonest and malicious willful dissemination of untrue statements.

2) Because this matter has already been thoroughly investigated and adjudicated via a formal research ethics inquiry, we need to alert the editors of PeerJ to the fact that this manuscript submitted by During, Voeten, and Ahlberg, which repeats these allegations, is defamatory and potentially libelous. In light of the severity and nature of the allegations that were leveled at the corresponding authors of DePalma et al., 2021 by During and Ahlberg in their complaint to UoM, and the fact that all of those allegations were overturned and found without merit, the additional attempts by During and Ahlberg to repeatedly circulate the same critique, including the draft submitted to PeerJ, are unethical, fail to honor the official process that had been undertaken, and do not follow a pattern of impartial and altruistic academic pursuit. They instead appear to constitute an abuse of the scholarly journal system for reasons other than scientific merit.

3) We also want the editors of PeerJ to be aware that this manuscript has been redundantly submitted to various journals by During and Ahlberg over the past two years and has already been published at least twice, see for example:

PubPeer (https://www.pubpeer.com/publications/9B9D041BD4D3633C2D4F99D002DF87),

PCI Paleo (DOI:10.24072/pci.paleo.100221)

These two versions of this manuscript are essentially the same as what has been submitted to
PeerJ, in fact the PCI Paleo manuscript is verbatim the same and was published online on the 26th of March, 2024, only ten days before the date of April 5th, 2024 assigned to the PeerJ submission.

Therefore the authors are attempting to re-publish the same material which has already been published and has an active DOI assigned. This does not meet the criterion for valid new research that would be suitable for publication. It is a repetition and a re-hash.

4) The manuscript was also previously submitted to Scientific Reports which rejected it on the grounds that it provided no scientific value, particularly in light of the fact that the topics discussed therein are already in the process of being either properly clarified or minor errata being corrected. The precedent set by the rejection, and its basis, should have bearing on the assessment of the suitability for publication of the same content that has now been submitted to PeerJ.

5) Lack of disclosure was also problematic. It is clear that the authors concealed from PeerJ the true extent to which their manuscript had previously been submitted, breaching a policy of transparency that is typically standard and required by journals. Under the Declarations heading of the PeerJ submission, which asks “Are any elements of this paper or text under consideration at any other journal, or have they been published elsewhere already?”, During, Voeten, & Ahlberg concealed that they had previously submitted the same material to Nature Scientific Reports, as well as its submission in other online formats, including PubPeer. During, Voeten, & Ahlberg concealed multiple occasions on which they distributed the same material, falsely implying that the content had not already been exhaustively belabored, and implying a greater novelty and applicability for the new submission. This misrepresentation to the journal, seemingly to make the manuscript appear more favorable for publication, is problematic.

6) In our experience, a critique would normally appear in the same scholarly journal as the paper that it critiques. The present attempt by During, Voeten, & Ahlberg to side-step that procedure and publish in a different journal is irregular, departs from standard practice, and should be regarded with concern.

7) In the PeerJ manuscript draft, there is mention of the During et al., 2022 manuscript and its relation to the DePalma et al., 2021 paper being critiqued. However, almost none of the coauthors of During et al., 2022, except for During and Ahlberg, have joined the PeerJ critique draft or any of the aforementioned near-identical iterations that were circulated as complaints. Abandonment by nearly all other coauthors in those efforts strongly suggests their lack of confidence in the content being presented and the course of action. This should also be taken into consideration.

For the seven above reasons, we stand by our first recommendation that the submitted manuscript is not a fit candidate for publication. However, that assessment is supported by multiple additional key facts related to the content of the manuscript. Below, following the format for PeerJ review, we address specific examples of concerns with the During, Voeten, & Ahlberg manuscript revised draft, and address their inappropriate use or otherwise unsuitability for publication.

[Basic Reporting]
Basic reporting was clear but misleading, as it circumvented known facts to arrive at unsupported conclusions. The authors lacked sufficient background in areas such as field work specifically related to this project and recognition of seasonal growth patterns in fish bone to adequately support their conclusions. The article was a belabored repeat of previous submissions that occurred after the issues raised in the draft were already adjudicated, and therefore does not constitute an example of professional scholarly work.

[Experimental Design]
This manuscript did not involve scientific experimentation. The questions raised were not relevant and were instead misleading as they were not reflective of the full facts that were known by the authors. Because of that, and because the draft has already been published multiple times elsewhere, it does not fall within the aims and scope of the journal. The aforementioned belaboring of topics that have already been addressed, their redundant prior circulation in public domain, and failure of the authors to disclose those facts do not follow a pattern of altruistic impartial scientific pursuit and therefore do not follow a high ethical standard. The manuscript is a highly biased and uninformed attempt to re-review a publication. The issues raised have been dealt with exhaustively elsewhere, and therefore it cannot truly be regarded as primary scientific research.

[Validity of the Findings]
Problematically, the information presented by During, Voeten, and Ahlberg here is redundant, misleading, and repeats what has already been belabored. All sections exhibit insufficient support of the conclusions or deal with topics that have already been addressed. This has not been sufficiently remedied since the first review, and the resubmitted draft has failed to address issues or inadequacies raised by the first review and Editor. Comments below are organized by their respective sections.

[Stable isotope records with conflicting migratory signals]
Section summary comments
This section makes broad claims about the viability of interpreted migration signals in fossil organisms without acknowledging that insufficient background data is available on migration or feeding habits of the animals in question, or acknowledging that isotopic signals that may indicate migration, if reliable and reflective of said migration patterns, are not central to any key argument or core interpretation of the paper. This section fails to adequately support the authors’ criticism against the use of isotopic data to resolve seasonal cyclicity in the acipenseriform bones. Annual cyclicity is expected to be recorded in the bones of animals that experience annual fluctuations in any condition that affects bone growth, whatever those variables happen to be. For purposes of tracking annual cyclicity, identifying which variable affected the bone growth is of minimal importance compared to the fact that an annual pattern exists. During, Voeten, & Ahlberg point to questions related to oxygen isotope shifts that were tentatively interpreted to indicate migration patterns in the sturgeon, despite those interpretations having no bearing on the conclusions of the DePalma et al., 2021 study. Because not all sturgeon taxa exhibit migratory behavior, one would expect that a migration signal, if reliable and not an anomalous isotopic signature or artifact linked to other factors, would be inconsistently present for sturgeon among the population, thereby providing limited to no utility in supporting the conclusions of DePalma et al., 2021. For that reason, the interpretation of migration was never a key point in the study and was not relied upon for the conclusions. During, Voeten, & Ahlberg base part of their critique on patterns in carbon isotopes between the paddlefish and sturgeon in context of their dietary practices, which is problematic because there exists no detailed body of knowledge on extinct Mesozoic acipenseriform fish diets or feeding practices. Indeed, the specific environmental and metabolic factors that influence the annual fluctuations of bone growth even of extant acipenseriforms are very poorly understood (“…it is not easy to explore the effects of environmental and metabolic variations recorded in the spines because the processes governing bio-mineralization and growth of these pieces are still poorly known”; Meunier 2002; Bakhshalizadeh wt al., 2017). Because the authors lacked sufficient background knowledge on the feeding practices of the extinct fish, they attempted to rely on feeding data from some modern fish communities instead, without the ability to demonstrate that they can be applied to the fossil taxa. For example, many factors have never been demonstrated for fossil Mesozoic paddlefish or sturgeon, including whether they fed in the topwaters or bottomwaters, consumed live food or passively consumed detritus, possessed similar feeding practices, or feeding practices that differed markedly, etc. During, Voeten, & Ahlberg fail to adequately uphold their criticism that the isotope data from DePalma et al., 2021 does not support the conclusions that the fish perished in the Spring-Summer paleo equivalent. In addition, we furthermore point out that the conclusions of the DePalma et al., 2021 study are additionally in no way dependent on the isotopic data and the conclusions are fully supported even in its absence. (UoM investigation finding: “the overall conclusions of the paper still stand should the stable isotope results be removed”). The incorrect statements made by During, Voeten, & Ahlberg in this section, which fail to support their conclusions, demonstrate insufficient background/experience in this field of work by the authors.

[Primary Data]
Section summary comments
During, Voeten, & Ahlberg state that no isotope data is provided in the DePalma et al., 2021 paper, however they misrepresent themselves by their omission of certain keys facts. For example, as with many other studies, the data was available upon request and was, in fact, immediately supplied to the journal when requested. It was also supplied to the UoM during their rigorous investigation of the allegations made by During, Voeten, & Ahlberg. Not only did During, Voeten, & Ahlberg know about that, but they, too, received a copy of the data. The concealment of those three facts from PeerJ during submission is problematic and demonstrates a biased presentation of incomplete facts to support or imply an improper conclusion. While the data was always available upon request, the journal may have overlooked their opportunity to add a written mention of that fact to the paper during final edits, but it was never withheld. To that end, even the title of During’s submitted manuscript is overtly misleading and incorrect, as instead of “Calibrations without raw data” (which she and the journal know is untrue) it is closer to “Calibrations without a concise data availability statement”. (UoM investigation conclusion: “’the low-resolution blurry photos of paper printouts’ provided by the Respondents to Scientific Reports and the Panel of Investigation…were consistent with what would be expected as a summary of calibrated data reported back to a client from a service isotope lab”). These data sheets nonetheless constituted scientific data by any definition. (Additional UoM conclusion: “there was independent evidence from 5 individuals (Oleinik, Burnham, Cichocki, Larson, Erikson) that DePalma’s isotope data pre-dated During’s visit to Tanis in 2017 and that one (Oleinik) had seen the plots in 2016 or early 2017 and confirmed they were the same ones that appear in the Scientific Reports paper”.

[Analytical Facility]
Section summary comments
During, Voeten, & Ahlberg state that some details related to the contributions by our late colleague are unclear due to his untimely decease. While regrettable, this, too, was previously surmounted in the UoM investigation, which was known by During, Voeten, & Ahlberg but not disclosed by them. Our late colleague facilitated the completion of various specialized tasks that were closer to his area of expertise than anyone else on the paper at that time. (UoM investigation conclusion: “although McKinney’s institution did not have the kind of apparatus supposedly used for the analysis, this was not evidence that McKinney had not sent the samples elsewhere for analysis”, in addition to the assessment that “’the low-resolution blurry photos of paper printouts’ provided by the Respondents to Scientific Reports and the Panel of Investigation…were consistent with what would be expected as a summary of calibrated data reported back to a client from a service isotope lab”.

[Methods]
Section summary comments
As mentioned regarding the analytical facility, the death of our colleague Curtis Mckinney occurred slightly before full completion of his contribution, which would have included his write-up of the isotopic methods. Using the notes and discussions that we had during the process, the methods were reconstructed as fully as possible, but some portions remained unknown. Our attempt was to compile methods that were as complete as those notes and discussions would allow, while not including any known falsehoods. This, too, was raised and surmounted in the UoM investigation, the process and outcomes of which are known to During, Voeten, & Ahlberg and which they failed to disclose.

[Sampling Density and Amount of Carbon]
Section summary comments
During, Voeten, & Ahlberg here fail to support their proposed conclusion that the sampling density is incompatible with the ability to retrieve sufficient sample for analysis. What they concealed from this section is that they are aware of multiple additional isotopic experts who were consulted and determined that it would in fact be possible to functionally retrieve sufficient sample. The description of drill bit shapes and sizes demonstrates an unfamiliarity with the process and technique, as a progressive inward-directed sampling along a peripheral transect is limited only by the incremental step-size capabilities of the micromill, and not the diameter of the burr. It is therefore completely unclear how exactly the number of samples per length of the sampling transect can “correspond”, as claimed by During et al. to the drill diameter. No such correlation exists. This, too, was raised and surmounted in the UoM investigation, the process and outcomes of which are known to During, Voeten, & Ahlberg and which they failed to disclose. The UoM investigation conclusion (which During, Voeten, & Ahlberg are aware of) further contradicted the claims by those authors in the present draft, establishing that “typically mass spectrometers require 25ug of carbonate for reliable analysis (approximately 500ug of fresh bone is required to reliably yield >20ug carbonate). However, it is just within the bounds of possibility for a skilled operator to recover the number of samples, and for a well-run isotope lab to recover the data”.

[Graphs in the Paper and Supplementary Materials]
Section summary comments
During, Voeten, & Ahlberg point to a variety of errata exhibited by the graphed isotopic data, however, more troublingly, they made no mention that every point raised in this section was laboriously addressed, discussed, explained, and adjudicated in the UoM investigation, all the details of which are known by During, Voeten, & Ahlberg. Their conscious and willful concealment misrepresents the facts of the issue and creates an intentional bias that otherwise could have been avoided. While the topics in this section have already been fully and thoroughly dealt with, it is worthwhile to point out some facts in response. For example, there do indeed exist a number of errata that ultimately result primarily from the untimely death of our colleague and the effort to organize his work, including the manual transcription of figures from his data sheets. The errata are largely applicable to graphs, graphed points, etc., that were manually transcribed from his printed data sheets as carefully as possible. (UoM investigation conclusion: “The inconsistencies in the data were explained as genuine errors resulting from the lack of raw data as a consequence of the death of McKinney and DePalma’s use of the interim data sheet to hand-draw the graphs”. Other criticisms noted in this section deal with factors that the authors (During, Voeten, & Ahlberg) evidently did not understand or fully read in DePalma et al., 2021 or the UoM investigation report. For example, as explained previously in the UoM investigation and known already to During, Voeten & Ahlberg, none of the graphs are identical. Because the specimens came from multiple animals with near identical life histories from a synchronous death assemblage, and in some instances multiple sets from the same individual, similarities in patterns are not only perfectly normal, but they are expected. Because the growth lines in fish bones are wavy and sinuous, they exhibit compressed or expanded representations of the same growth band pattern, for example in troughs as opposed to lobes. So, in one individual, a single sliced surface can contain a 5-band pattern of growth that is perhaps 3 mm thick in one expanded region or 0.3 mm thick in an adjacent compressed region, with every variation in between. Another example that During, Voeten, & Ahlberg cite is the use of data point icons in the supplemental materials that they misinterpreted as error bars because they did not read the actual figure caption that explicitly states so. Another example is the mention of one specimen number that is listed twice, which was previously clearly explained, and demonstrated during the UoM investigation as being correct, because it reflected multiple sampling of the same specimen, however During, Voeten, & Ahlberg concealed that fact, implying a discrepancy that pointed toward lack of integrity. (UoM investigation comment: “The Appeal Panel requested clarification from DePalma who explained that Fig 2 was a repeat analysis of the same sample, and provided the data from which it was plotted”. At the time of their submission to PeerJ, During, Voeten, & Ahlberg knew this, and every single other detail in their section about the graphs, which they could have transparently revealed to the journal but instead they only mentioned their initial criticism and omitted everything that transpired since then in the process of addressing and/or satisfying it. As we had previously mentioned during the UoM investigation, of which During, Voeten, & Ahlberg are aware, the legitimate errata in the graphs that are linked to the manual transcription of the data resulted in near imperceptible shifts that were not sufficient to affect the conclusions of the study in any way whatsoever. (UoM investigation conclusion: “the differences between curves derived from the numerical values in the tables and the curves published in the Scientific Reports paper were minor, as confirmed by two independent individuals (Oleinik and Smit – During’s MSc advisor) and were likely the result of the published curves being copied from original plots”. The UoM investigation upheld the assessment that data fabrication/manipulation had not occurred, even after an appeal process initiated by During & Ahlberg, and went on to comment that “the overall conclusions of the paper still stand should the stable isotope results be removed”.

[Thin sections in the Supplementary Materials]
Section summary comments
In this section, During, Voeten, & Ahlberg claim that DePalma et al. manipulated an image in the supplementary section, flipping an image that had been photographed twice from the same side. Not only is this statement false and insufficiently supported, but the UoM investigation thoroughly looked into it with multiple experts, concluding that manipulation had not taken place, that the images were not both of the same face of the slide, and that the claim was unfounded. Furthermore, During was cautioned by the UoM Chair of Ethics, that, in light of During, Voeten, & Ahlberg being aware that their claim was false, if they were to perpetuate that claim they would be knowingly and intentionally circulating false and misrepresented facts in a malicious way.

[Fish sizes]
Section summary comments

During, Voeten, & Ahlberg claim that During observed no juvenile or sub-yearling fish while she was on-site, implying that none were there. This statement, at best, indicates that they did not observe or recall the sub-yearling acipenseriform fish skull included in Figure 1 of DePalma et al., 2021 as a Micro-XRF map. At worst, the statement reflects a willing concealment of their knowledge of sub-yearling fish fossils at the site, including the personal experience of During. In addition, if During did not observe any sub-yearling fish during her brief site visit, that does not indicate that the fish were not present, but rather is reflective of During’s inexperience working or identifying fossils in that field setting. During lacks experience with field work in the Hell Creek Formation (we were told that her ~10-day site visit in 2017 was her first experience in the Hell Creek Formation and she had accrued fewer than two week’s additional field time there in the 7 years since then, as far as we are aware). As a result, it was a challenge for During to detect most fossil material and it had to be shown to her on many occasions. She even personally encountered sub-yearling fish herself, although not intentionally- the well-preserved fins and partial body of what would have been a ~14 cm juvenile acipenseriform fish were discovered freshly broken in her debris pile, unintentionally destroyed by During as the fossil went unnoticed while she dug into the outcrop [FIG 1]. While During may not have measured any fish lengths during her short visit, other researchers on-site did. During, Voeten, & Ahlberg claim that the smallest fish at the site are all 15 cm long, and in support of that they erroneously cited a graph from another publication that intentionally began its tally at the 15 cm size range because the densely tangled mass-death assemblage made it problematic to accurately tally fish smaller than that in situ prior to preparation of the blocks. Regarding growth estimates, the comparison with the seasonal growth ranges of modern fish involved clearly citing multiple compiled ichthyological works, contrary to claims by During, Voeten, & Ahlberg.

[Conclusion of a Spring Death]
Section summary comments
In this section, During, Voeten, & Ahlberg state that they did not clearly observe osteocytes in the fish bone images, and therefore conclude that discerning the season of death would be challenging or impossible. This claim is perplexing, clearly demonstrates their unfamiliarity with the subject material, and demonstrates their inability to sufficiently support their claim. The assessment of seasonal oscillation in the bones of fish (including acipenseriforms) is routinely, reliably, and most-frequently carried out via counting the annuli, or growth bands, in the bone cross-sections (Kolhorst et al., 1980; Brennan & Cailliet, 1989; Jackson et al., 2007; Bakhshalizadeh et al., 2011; Neely & Lynott, 2016; FIG 2A). Each year is represented by a couplet comprised of a light and dark band, with the denser, dark bands grown during the favorable growth period (Spring-Summer) and the lighter, more-translucent, less-dense bands grown during the unfavorable growth period (Fall-Winter). In their response, During, Voeten, & Ahlberg yet again attempt to shift focus to osteocyte density when that method is by far not standard and is peripheral to the more widely used method of examining the dual bands of the annuli. During’s comment that osteocyte density is more commonly used than examination of growth bands is incorrect. To suggest that the seasonal assessment should be based on osteocyte distribution is peculiar, uncustomary, not requisite, and not aligned with standard practice. Our assessment, based on the very clear patterns exhibited by strongly defined growth band couplets that are on par with modern counterparts [FIG 2] is robust, unambiguous, and was favorably critiqued by multiple professional bone histologists prior to publication of DePalma et al., 2021 (and listed in the acknowledgements thereof). Based on studies of annual growth in modern acipenseriformes, the bone histological data, alone, was sufficient as the exclusive indicator of season-of-death for DePalma et al., 2021, barring all other evidence presented by that study. In addition, During, Voeten, & Ahlberg attempt to undermine the conclusions of DePalma et al., 2021 by claiming that the data indicates a Summer death rather than Spring. While we disagree with the focused exclusivity of that assessment, their comment further indicates their unfamiliarity with the DePalma et al., 2021 paper, which interpreted a Spring-Summer range for time of death, which is consistent both with the data of DePalma et al., 2021 and During’s own analysis of the process.

[Conclusions]
Section summary comments
During, Voeten, & Ahlberg state that “The stable isotope graphs presented in DePalma et al.'s paper, as depicted in Figure 25 and the Supplementary Materials, exhibit patterns that deviate from what would be expected from direct analytical outputs”. With that statement, During, Voeten & Ahlberg misrepresent themselves by concealing that they already knew the reasons for minor errata in the graphs, as they had already been thoroughly addressed during the UoM investigation. They also cite absence of raw data, when they already had in their hands the raw data, which was immediately provided upon request, as is customary for any scientific paper that does not contain it in the text or supplement. They claim that the evidence presented does not align with the assertion/conclusions of our study, however that statement is incorrect and fully unsupported, because the multiple, independent, mutually reinforcing lines of evidence in DePalma et al., 2021 robustly support the conclusions of a Spring-Summer time of death. During, Voeten, & Ahlberg claim that the osteohistological slides, fish growth, and isotope data fail to indicate the season of death, however those are all thoroughly discussed and explained in the DePalma et al., 2021 paper, including graphical explanations of the histological subdivisions laid down during favorable vs lean growth periods and showing that the fish died at some point during the favorable period (Spring-Summer). That assessment was also made by the UoM investigation, which found that “the overall conclusions of the paper still stand” even in absence of the isotopic data. system for personal reasons. As such, the manuscript of During, Voeten, & Ahlberg is not fit for publication and would not serve any useful benefit if that were to be published. Regardless, it can already be viewed via multiple other hyperlinks, removing any justification for adding yet another.

[References]
The Reference portion of the manuscript submitted to PeerJ by During et al, with only 31 items in it, contained an error indicative of unprofessional editing of the manuscript. References #20 (Lines 320-323) and #22 (Lines 329 – 332) are identical and simply redundant, which have now been corrected by the authors.

>>>>>>>>>>>>>>>>>
Response to During’s editorial comments

During: Blue
DePalma: Red
Editor comment: [Basic reporting] The main points are clearly and concisely stated (compare reviewer 1). Some more supporting references should be given to support the recognition of seasonal growth patterns (compare reviewer 2: see further point).

During response: Reviewer 2 correctly notes that dark and light bands have traditionally been used to study annual growth patterns. However, it is now more common to employ osteohistology for a more precise understanding of growth, as the binary nature of dark and light bands can be limiting.

DePalma comment to During response:This is incorrect and misleading, as the traditional and most widely used approach is to examine the peak-season/lean-season growth bands. During confirms what I stated in my previous review- that the use of osteocytes is a peripheral technique.

During response: To better explain this, we have included two additional references that specifically explain osteohistology and the expressions and significance of osteocyte patterns in endochondral bone. This approach underlies our analyses, but is notably unavailable for the depicted thin sections published in DePalma et al. (2021) due to their low image resolution.
DePalma comment to During response: This too is incorrect, as the images used in DePalma et al., 2021 are in fact as clear if not clearer than the histological sections used to age modern fish (see attached image, Figure 2). Notably, the growth bands exhibited by DePalma et al., 2021 unanimously indicate a Spring/Summer time of death, supporting the conclusions of all mutually reinforcing lines of evidence in that paper.

Editor comment: [Experimental design] Not applicable or needed as it concerns a reply to a published paper using as much publicly available data as possible (compare reviewer 1).

During response: We thank the editor and reviewer 1 for confirming this.

Editor comment: [Validity of the findings] The authors disclosed the information about previous publication and peer-review through PCI and falls with the scope of the journal as it would otherwise not be sent out for review and considered as a reply if it was not.

During response: We thank the editor for confirming this.

DePalma comment to During response: The authors withheld, however, the other instances in which they circulated near-copies of the same manuscript, thereby concealing the extent to which it has circulated and implying more novelty or utility for the PeerJ submission.

Editor comment: [Stable isotope records with conflicting migratory signals] It should be more clearly stated that that points are not central to the core interpretation of the original paper (compare reviewer 2) but if they are stated in the original paper, they can be entertained in the rebuttal. Please add more clearly that you cannot rule out some points due to insufficient background data is available on migration or feeding habits of the animals in question or cite such information if available. Clearly state you assume similar principles in extinct fish feeding if this is the case and back it up with references. Add information which of conclusions of the original paper (e.g., Spring- Summer death?) would still stand if isotopic data is discarded.

During response: We do not suggest that all our findings pertain to the core interpretation of the DePalma et al. paper and appreciate the confirmation that accessory claims in that paper are equally subject to scrutiny. We also feel that any novel claim made in a research paper should have sufficiently support from academic literature. We agree that supportive evidence is required to confirm or reject the interpretation by DePalma et al. (2021), and feel that exactly that is missing from DePalma et al. (2021) to begin with. Because the stable isotope values of Tanis fossils have only recently been produced and studied, background data is understandably limited. Modern fishes may offer some context, but do crucially not represent the exact same taxa, ecological strategies, and prey composition as the Tanis piscifauna, and suffer from contemporary migratory restrictions that will variably influence dietary habits as well.
We here simply wish to flag an unexpected correlation in the DePalma paper that, without additional context or explanation, raises strong doubts. However, since background information is understandably scarce for this specific topic, we have rewritten the last sentence of the referred paragraph to now say:
“While the observed

---

## Round 0.3 · accepted · Accept

Thank you for addressing the outstanding issues that were raised by me as Editor, and those by the previous Editor of your manuscript during the last round of review. I also appreciate your careful attention to outstanding issues that were raised by reviewer #2, noting that many of those relate to outcomes or information not available in the public domain. On consultation with the PeerJ Paleontology Section Editors, including the opinion of an additional Section Editor who evaluated the overall process, we consider your revision addresses our previous concerns and your manuscript should be accepted.